# BudgetThinker: Empowering Budget-Aware LLM Reasoning with Control Tokens

## Abstract

Recent advancements in Large Language Models (LLMs) have leveraged increased test-time computation to enhance reasoning capabilities, a strategy that, while effective, incurs significant latency and resource costs, limiting their applicability in real-world time-constrained or cost-sensitive scenarios. This paper introduces BudgetThinker, a novel framework designed to empower LLMs with budget-aware reasoning, enabling precise control over the length of their thought processes. We propose a methodology that periodically inserts special control tokens during inference to continuously inform the model of its remaining token budget. This approach is coupled with a comprehensive two-stage training pipeline, beginning with Supervised Fine-Tuning (SFT) to familiarize the model with budget constraints, followed by a curriculum-based Reinforcement Learning (RL) phase that utilizes a length-aware reward function to optimize for both accuracy and budget adherence. We demonstrate that BudgetThinker significantly surpasses strong baselines in maintaining performance across a variety of reasoning budgets on challenging mathematical benchmarks and multimodal reasoning benchmarks. Our method provides a scalable and effective solution for developing efficient and controllable LLM reasoning, making advanced models more practical for deployment in resource-constrained and real-time environments.

## 1 Introduction

A recent breakthrough in Large Language Models (LLMs) is the paradigm of test-time scaling, where models are prompted to "think longer" by generating extended Chain-of-Thought (CoT) reasoning before producing a final answer (DeepSeek-AI, 2025; OpenAI, 2024). This approach has achieved state-of-the-art performance on complex reasoning tasks in domains such as advanced mathematics (DeepSeek-AI, 2025; Google, 2025), code generation (Hui et al., 2024; Yang et al., 2025), and multimodal reasoning (Team et al., 2025a; Huang et al., 2025; Shen et al., 2025). However, these performance gains come at the steep cost of increased latency and computational overhead, making these reasoning models impractical for real-world systems with strict performance constraints, such as latency-critical agents and robotics (Sui et al., 2025).

This fundamental trade-off between reasoning quality and computational efficiency highlights a critical need for *controllable CoT reasoning*—the ability for a reasoning LLM to dynamically adapt its reasoning process to conclude within a specified budget. Such a budget may be a hard latency limit imposed by an application, such as the sub-100 ms requirement for autonomous driving (Lin et al., 2018; Chen et al., 2025a), or a flexible constraint based on user tolerance for cost and delay in agentic systems. Furthermore, the variability of computational resources in real-world deployments (*e.g.*, diverse hardware, dynamic server loads) necessitates that models can precisely tailor the length of their output to fully utilize the available budget (Cai et al., 2020; Yu et al., 2019). We term this crucial instruction-following capability **budget adherence**.

However, existing approaches to shortening reasoning length have proven insufficient for budget adherence. Directly adding length control instructions into the prompt (Pu et al., 2025; Sun et al., 2025; Xu et al., 2025b) often fails to reliably control output length (Han et al., 2024). Some LLM post-training methods encourage models to switch between reasoning modes like "thinking" or "non-thinking", or to generate responses based on reasoning efforts (Yang et al., 2025; OpenAI, 2025b;a; Team et al., 2025d; Jiang et al., 2025; Team et al., 2025a). However, these approaches still lack the

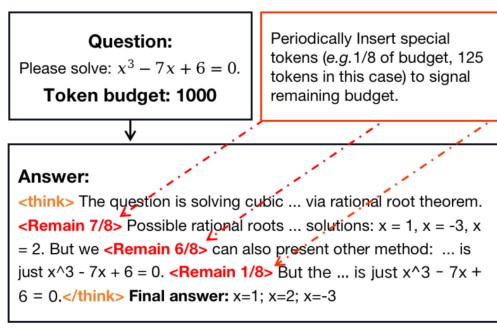

(a) Example of budget-aware answer generation with Control Tokens.

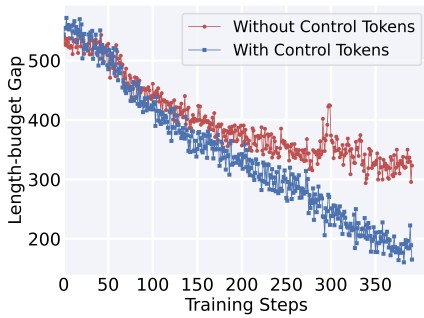

(b) Effect of Control Tokens on reducing length–budget gap.

Figure 1: Illustration of the BudgetThinker framework and the impact of Control Tokens. (a) An example of budget-aware generation, showing how BudgetThinker inserts control tokens into its reasoning process to stay aware of the remaining token budget. (b) Effect of Control Tokens on training: The gap between the generated answer length and the target budget during reinforcement learning.

fine-grained control necessary for variable budgets, where the target is a fixed number set by the user or system, not a vague description of a reasoning level. Even recent research introducing training pipelines with Supervised Fine-Tuning (SFT) and Reinforcement Learning (RL) (Hou et al., 2025; Aggarwal & Welleck, 2025; Wu et al., 2025b; Liu et al., 2025; Xia et al., 2025; Qu et al., 2025; Yu et al., 2025c) demonstrates insufficient capability. As illustrated in Figure 1b, the gap between the LLM-generated length and the target budget remains large (around 300 tokens) throughout training, highlighting a persistent challenge.

In this paper, we address the challenge of precisely controlling the length of a model's thought process. We argue that merely stating a budget constraint in the initial prompt is insufficient, and the model needs to be continuously reminded of its remaining token budget as it generates its response. To achieve this, we develop a novel training and inference framework that enables precise control over the length of the CoT. Our method periodically inserts special *Control Tokens* that act as explicit signals of the remaining budget, as illustrated in Figure 1a. As shown in Figure 1b, training with these control tokens leads to a faster and more stable reduction in the gap between the generated length and the target budget, demonstrating their effectiveness in stimulating budget adherence.

We evaluate our method based on DeepSeek distilled Qwen 2.5 1.5B, 7B (DeepSeek-AI, 2025) and RoboBrain-7B-2.0 (Team et al., 2025a), on challenging reasoning benchmarks including MATH-500 (Hendrycks et al., 2021), AMC 2023, Embodied Reasoning QA Evaluation Dataset (EQRA) (Team et al., 2025b), and Spatial Aptitude Training Benchmark (SAT Benchmark) (Ray et al., 2025). Compared to state-of-the-art efficient reasoning methods (Hou et al., 2025; Liu et al., 2025; Xu et al., 2025b; Sun et al., 2025) and original models, our approach demonstrates a better trade-off between accuracy and token budget. Specifically, our method improves accuracy by an average of 4.9% across all tested budgets and exhibits more precise adherence to the specified constraints.

Our main contributions are:

- We introduce a method to achieve controllable output length in LLMs by inserting special tokens during the inference process.
- We design a comprehensive training process, progressing from SFT to RL, which incorporates a length-aware reward function. This allows the model to adapt to control via special tokens and produce higher-quality, budget-aware outputs. Our method is designed as a plug-and-play module that complements existing test-time scaling training procedures.
- Across multiple foundational models (including both LLMs and MLLMs), we demonstrate that our approach surpasses strong baselines and datasets under various reasoning budgets.

## 2 SPECIAL-TOKEN CONDITIONED BUDGET-AWARE REASONING

We present BudgetThinker, a budget-aware generation framework that guides Large Language Models (LLMs) to adhere to a predefined token budget. Our method achieves this by conditioning the model on explicit control signals throughout the generation process. We enhance budget adherence by (i) introducing a dynamic control token insertion strategy that continuously informs the model of the remaining budget and (ii) developing a two-stage training pipeline that combines Supervised Fine-Tuning (SFT) and Reinforcement Learning (RL) to teach the model to effectively interpret and respond to these signals.

### 2.1 SPECIAL-TOKEN BUDGET SIGNALING

Our method enforces a target output token budget by augmenting the standard autoregressive generation process. The core idea is to insert a small, fixed set of special control tokens into the sequence at positions corresponding to **fractions of the total budget**. These tokens act as explicit signals to the model, indicating how much of its budget has been consumed. Let $B$ be the target budget and $K$ be a fixed hyperparameter for the number of control signals. We introduce a set of $K$ special control tokens, $\mathcal{C} = \{c_1, c_2, \ldots, c_K\}$, where $c_k$ signals that the generation has progressed through the $k$-th fraction of the total budget (*e.g.*, $c_1$ signifies 10% completion if $K$=10). During generation, the token $y_t$ at timestep $t$ is determined by the following piecewise function:

$$y_t = \begin{cases} c_k & \text{if } t = k \cdot \lfloor B/K \rfloor \text{ for an integer } k \in \{1, \ldots, K\} \\ y'_t \mid (y'_t \sim \pi_\theta(\cdot|x, y_{<t})) & \text{otherwise} \end{cases}$$

At each timestep $t$ that corresponds to the end of a budget fraction $k \cdot \lfloor B/K \rfloor$, we deterministically insert the corresponding control token $c_k$. For all other timesteps, the next token $y'_t$ is sampled from the LLM's distribution $\pi_\theta$, conditioned on the full history of both model-generated tokens and our inserted control tokens.

This **ratio-based signaling** is highly scalable. In contrast, a naive baseline like Fixed-Interval Signaling—which inserts a token every $I$ steps-would require a potentially unbounded number of unique control tokens as the budget $B$ increases. Our approach always uses the same fixed set of $K$ tokens, making it robust and efficient across widely varying budgets.

### 2.2 SFT DATA CURATION

To teach the LLM the semantics of our control tokens, we construct a specialized dataset for Supervised Fine-Tuning (SFT). The curation process involves two key steps: budget assignment and dataset balancing.

**Budget Assignment.** First, for each original data sample $(x, y)$, we assign a budget $B$ by rounding the answer's length, $|y|$, up to the nearest multiple of a granularity parameter $T$:

$$B = T \cdot \left\lceil \frac{|y|}{T} \right\rceil,$$

This calculation method is intentional. By design, the true answer length $|y|$ is almost always less than the allocated budget $|B|$. This gap teaches the model to robustly terminate its reasoning and stop generating tokens once the answer is complete, rather than artificially padding the output to fill the entire budget. Next, we transform each sample in the dataset. The input prompt is modified to include the budget constraint, and the target output is reconstructed to include the control tokens:

$$\hat{x} = \text{Concat}(x, \text{Please answer within } B_i \text{ tokens})$$

$$\hat{y} = \text{Insert}(y, \{c_j\}_{j=1}^K)$$

Here, *Insert* function adds the control token $c_{k+1}$ at each position $t$ when $t = k \cdot I$. Then, we get a reconstructed dataset $\{(\hat{x}, \hat{y})\}$, which we use to fine-tune LLM.

## 2.3 Dataset Construction and Balancing

To ensure the model generalizes across a wide range of output lengths, our SFT dataset is a balanced mixture of three distinct data types:

**(1) Long Reasoning Data:** Full-length CoT reasoning sequences from powerful teacher models, such as DeepSeek-R1 (DeepSeek-AI, 2025). **(2) Iteratively Compressed Reasoning Data:** To populate intermediate lengths, we use an LLM-based compressor to iteratively shorten the long CoT sequences (Kang et al., 2025). Critically, this compression is applied iteratively: the result of one compression step becomes the input for the next. This process continues until a comprehensive spectrum of shortened reasoning chains, derived from an original long CoT, is produced. This approach ensures our model learns to adhere to budgets requiring diverse levels of reasoning abbreviation. **(3) Non-thinking Data:** Direct, concise answers to handle scenarios with very tight budget constraints where elaborate reasoning is not feasible. This balanced composition is pivotal for developing a versatile and robustly budget-adherent model.

## 2.4 Reinforcement Learning

Following SFT, we employ Group Relative Policy Optimization (GRPO) algorithm (Shao et al., 2024) to train the model to generate responses that adhere to a specific token budget, $B$.

**Reward Design.** We design a composite reward function that balances factual accuracy with a penalty for deviating from the target length. For a generated response $y$, with ground truth $y_{gold}$ and budget $B$, the reward $R$ is defined as:

$$R(y, y_{gold}, B) == k_1 \cdot \mathbf{1}\{y = y_{\text{gold}}\} + k_2 \cdot \mathbf{1}\{\text{format}(y)\} + k_3 \cdot \underbrace{\max(1 - \gamma \cdot \left( \frac{B - |y|}{B} \right)^2, 0)}_{\text{Length Reward}}$$

Indicator functions $\mathbf{1}\{y = y_{\text{gold}}\}$ and $\mathbf{1}\{\text{format}(y)\}$ are the original correctness and format check of GRPO, and $k_1, k_2, k_3$ are coefficients that control the relative importance of each part. The *Length Reward* incentivizes the model to use the budget efficiently. It is based on *normalized length deviation*, $\delta = \frac{||y| - B|}{B}$, which measures the fractional difference between the generated length $|y|$ and the target budget $B$. This deviation is modulated by an *asymmetric penalty coefficient*, $\gamma$:

$$\gamma = \begin{cases} 1 & \text{if } |y| \leq B \\ r & \text{if } |y| > B \end{cases}$$

Here, $r > 1$ is a hyperparameter that imposes a larger penalty for exceeding the budget than for falling short of it. This design encourages the model to generate responses that are not only correct but also precisely tailored to the length constraint.

**Special Token Conditioned Sampling.** Our framework's special token insertion is integrated directly into the GRPO sampling loop. For each training question, BudgetThinker automatically injects control tokens at the appropriate budget fractions during generation. To provide a clear learning signal, the model completes its full reasoning rollout to calculate the length-based reward accurately. However, for the final answer extraction step, the reasoning trace is truncated at budget $B$. This allows the model to learn from the consequences of its full generation while still enforcing the hard budget constraint in practice.

**Mixed Budget Reinforcement Learning.** To build robustness across budgets, RL proceeds in two stages: (1) *Curriculum Stage*: Our LLM is trained iteratively on $k$ different token budgets. The curriculum proceeds by decreasing the budget across stages, following the sequence $B_1 > B_2 > \cdots > B_n$. (2) *Randomized Stage*: Each Training batch samples a budget $B_k$ uniformly from $B_{j\,j=1}^{\,n}$. This mixed strategy prevents catastrophic forgetting and maintains the model's proficiency across all learned budget levels.

## 3 EXPERIMENTS

In this section, we present a comprehensive empirical evaluation of BudgetThinker. We investigate its performance under various computational budgets and analyze its ability to adhere to these budgets (Section 3.2). We then analyze the impact of different control token insertion strategies (Section 3.3), iterative training (Section 3.4), and reinforcement learning of BudgetThinker (Section 3.5).

### 3.1 EXPERIMENT SETUP

**LLM Training Details.** We use DeepSeek-R1 (DeepSeek-AI, 2025) distilled Qwen-2.5-1.5B and Qwen-2.5-7B (Hui et al., 2024) as backbone models. For SFT, we create a dataset of 41k instances for text-based LLMs. The details are listed in Appendix A.2. We train all SFT models for 6 epochs with an initial learning rate of 2e-5 and a maximum context length of 12000.

For Reinforcement Learning (RL), we employ the Group Relative Policy Optimization (GRPO) algorithm (Shao et al., 2024). Models are trained for 10 epochs with GRPO hyperparameters $\alpha$ and $\beta$ set to 0.01 and 0.01, respectively. The maximum context length during RL is 10000.

We configure BudgetThinker by setting the number of control intervals to $K = 8$. For the iterative training stage, we start from a budget of 6000, then reducing it to 4000, 3000, and finally 2000, with 1 epoch for each budget. In the randomized training stage, we randomly sampling $B \in \{6000, 4000, 3000, 2000\}$ for 1 epoch. Further training details are shown in Appendix A.5.

Namely, the length reward will decade to 0 when the length of answer $y$ exceeds the budget $B$ by 1/4.

**MLLM Training Details.** We use RoboBrain-7B-2.0 (Team et al., 2025a) as the backbone model. For SFT, we use the visual question answering (VQA) data with CoT from RoboBrain-2.0, consisting of 37k question-answer pairs. In RoboBrain-2.0, the CoT data is generated from GPT-4o (Hurst et al., 2024), the lenght of which concentrates in 200-300 tokens. Therefore, we use the compressor based on GPT-4o to compress the CoT length for 3 iterations, creating a dataset of 111k.

For RL, we randomly sample 2k instances from the VQA training dataset of RoboBrain-2.0, train the MLLM for 6 epochs, with the initial learning rate of 1e-6. Specifically, in the We set the number of control intervals to $K = 3$ because the shorter reasoning length on MLLMs, and the training budgets $B \in \{200, 150, 100, 80, 50\}$ for the iterative training phase, and train 1 epoch on each budget. Then, we train 1 epoch for mixed budget stage. We remain the other hyper-parameters same with LLMs.

For fair comparison, we also train the original RoboBrain-7B-2.0 through the same procedure from SFT to RL using the same data and settings, with the difference that we don't add budget constraints during training and set the maximum context length to 500, which is far beyond the largest budget 200 in our experiments.

**Evaluation Details.** We evaluate our methods and baselines on 2 math benchmarks and 2 multimodal reasoning benchmarks: MATH-500 (Hendrycks et al., 2021), AMC 2023, ERQA (Team et al., 2025b), and SAT (Ray et al., 2025). To assess budget control, we evaluate performance across a range of reasoning budgets, from $B = 50$ to $B = 10000$ tokens. For inference, we utilize a modified version of the vLLM engine (Kwon et al., 2023). Our modified engine enforces the budget by truncating the reasoning process once the length limit is reached and appending the "`</think>**Final Answer**`" tags to signal the model for a final answer. We then allocate an additional 50 tokens for the final answer generation before terminating the process entirely.

**Baselines.** We benchmark BudgetThinker against five baselines: **(1) ThinkPrune** (Hou et al., 2025): A typical efficient reasoning framework that also uses GRPO for budget control but does not incorporate explicit control tokens. As the official ThinkPrune repository does not provide a 7B model, we limit our comparison to the 1.5B model scale. **(2) Laser** (Liu et al., 2025): A state-of-the-art efficient reasoning framework enhances both fast and slow thinking capability. **(3) Chain of Draft** (Xu et al., 2025b): A prompt based method that faciliates fast reasoning by prompting LLMs to draft concise intermediate thought. **(4) Early Exit**: (Sun et al., 2025) A token insertion based method that interrupts LLMs before budget and forces it to output the final answer. **(5) Original**

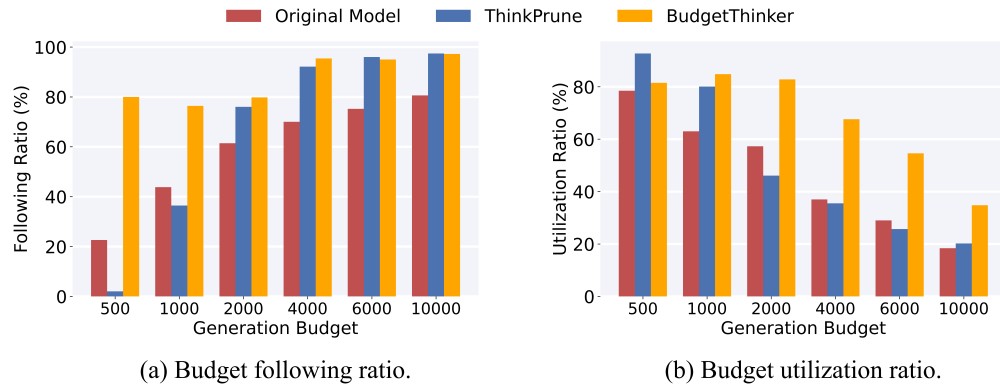

(a) Budget following ratio.

(b) Budget utilization ratio.

Figure 2: Budget following and utilization analysis of BudgetThinker-1.5B, ThinkPrune-1.5B, and the R1-1.5B on MATH-500. (a): The percentage of generated responses that terminate naturally within the specified budget limit ($B$). (b): For responses that finish within the budget, this shows the average generated length as a percentage of the total allocated budget ($|y|/B$).

**Models**: The DeepSeek-R1 distilled Qwen-2.5-1.5B and 7B models that serve as the foundation for our fine-tuning.

## 3.2 ACCURACY-BUDGET TRADEOFF

| | | | | | | | | | | | |
|---|---|---|---|---|---|---|---|---|---|---|---|
| *1.5B LLMs* | | | | | | | | | | | |
| **Method** | **MATH-500** | | | | | | **AMC** | | | | |
| | **0.5K** | **1K** | **2K** | **4K** | **6K** | **10K** | **2K** | **4K** | **6K** | **10K** | **30K** |
| ThinkPrune | 44.0 | 69.4 | 79.4 | 82.8 | 83.3 | 83.6 | **65.5** | 70.1 | 71.9 | 72.0 | 73.6 |
| Laser | 48.4 | 70.2 | 80.2 | 82.6 | 83.8 | 83.2 | 59.0 | 67.8 | 73.4 | **75.9** | **75.9** |
| Original Model | 20.4 | 45.8 | 63.0 | 70.0 | 69.2 | 70.2 | 36.9 | 46.9 | 55.6 | 61.6 | 67.8 |
| c2f | 56.8 | 70.0 | 77.4 | 80.8 | 81.8 | 80.0 | 54.4 | 65.9 | 62.5 | 64.7 | 68.3 |
| CoD | 38.0 | 59.4 | 64.2 | 64.8 | 65.2 | 65.2 | 33.4 | 46.9 | 48.1 | 49.4 | 66.1 |
| BudgetThinker | **59.4** | **75.0** | **81.6** | **84.8** | **85.8** | **87.4** | **65.5** | **70.4** | **74.0** | 75.0 | **75.9** |
| *7B LLMs* | | | | | | | | | | | |
| **Method** | **MATH-500** | | | | | | **AMC** | | | | |
| | **0.5K** | **1K** | **2K** | **4K** | **6K** | **10K** | **2K** | **4K** | **6K** | **10K** | **30K** |
| Laser | 53.8 | 80.0 | **88.8** | **90.6** | 92.0 | 92.2 | 67.8 | **83.8** | 86.3 | **87.5** | 88.1 |
| Original Model | 22.2 | 44.4 | 67.4 | 78.8 | 84.0 | 86.0 | 38.8 | 58.8 | 73.1 | 78.4 | 79.3 |
| c2f | 59.8 | 76.4 | 84.0 | 89.6 | 91.0 | 90.2 | 62.8 | 76.6 | 87.2 | 87.2 | 87.3 |
| CoD | 31.6 | 57.8 | 73.4 | 81.6 | 84.6 | 85.8 | 47.8 | 69.1 | 74.7 | 79.1 | 79.8 |
| BudgetThinker | 69.4 | **77.0** | 86.0 | 90.0 | **92.2** | **93.0** | **69.7** | 77.5 | **87.5** | **87.5** | **90.0** |
| *7B MLLMs* | | | | | | | | | | | |
| **Method** | **ERQA** | | | | | | **SAT** | | | | |
| | **50** | **80** | **100** | **150** | **200** | | **50** | **80** | **100** | **150** | **200** |
| c2f | 4.5 | 6.5 | 10.0 | 24.6 | 39.6 | | 23.0 | 55.0 | 66.0 | 69.0 | 74.6 |
| Laser | 37.1 | 38.8 | 39.6 | 40.4 | 42.3 | | 69.7 | 70.0 | 71.0 | 73.3 | 73.3 |
| BudgetThinker | **41.1** | **41.9** | **43.6** | **43.4** | **43.6** | | **70.7** | **72.0** | **72.7** | **74.0** | **76.7** |

Table 1: Pass@1 accuracy of BudgetThinker vs. baselines across various generation budgets. The numeric headers (*e.g.*, **0.5K**, ..., **30K**, **50**, ..., **80**) represent the Generation Budget.

We first evaluate the accuracy of BudgetThinker against the baselines across various token budgets, with results shown in Table 1. For results of MATH-500 and AMC 2023, BudgetThinker outperforms the original model and ThinkPrune across most budget allocations. Specifically, Bud-

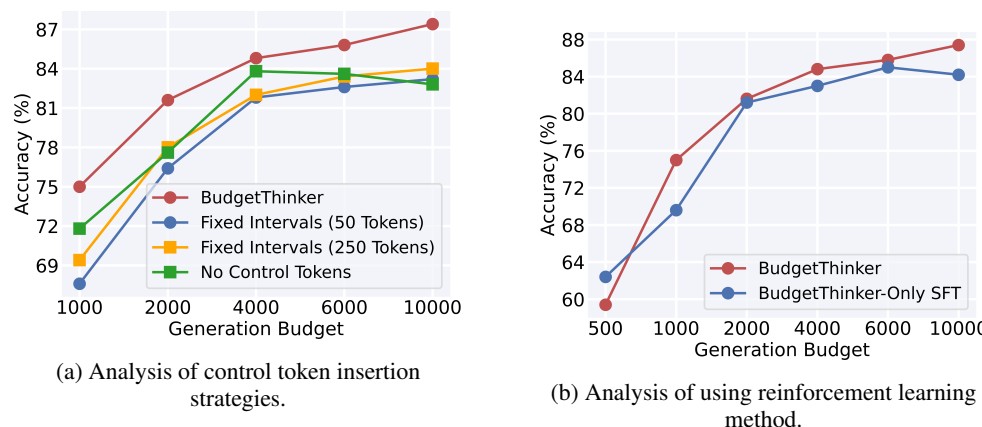

(a) Analysis of control token insertion strategies.

(b) Analysis of using reinforcement learning method.

Figure 3: Ablation studies on control strategy and RL training (MATH-500, 1.5B Model).

getThinker improves accuracy by 2.1% over Laser, 3.2% over ThinkPrune and 22.4% over Original model on average.

The better accuracy of BudgetThinker comes from the better understanding and utilization of budget, which is shown in Figure 2. We calculate the average relative length at each budget ($|y|/B$) for every answer that is within budget, which shows the capability for LLM to understand and fully use budgets. The original model, lacking any budget-specific training, exhibits poor capability to follow the specified limits. While ThinkPrune (Hou et al., 2025) shows improved budget awareness, it often underutilizes the allocated budget, prematurely concluding its reasoning process, which can negatively impact performance on complex problems. In contrast, BudgetThinker demonstrates superior budget adherence. Guided by the remaining budget tokens, it effectively utilizes the entire allocated length without significant overruns. This precise control allows it to achieve a better balance between computational cost and accuracy, leading to its state-of-the-art performance. We offer a detailed analysis in Appendix A.7.

## 3.3 ANALYSIS OF CONTROL TOKEN INSERTION STRATEGIES

We analyze the performance of BudgetThinker under four distinct control token insertion strategies to understand their impact:

- **No Control Tokens:** An ablation where no special tokens are inserted, relying solely on the RL objective for budget control.
- **Fixed Interval (50 tokens):** Control tokens are inserted at a fixed, frequent interval of every 50 tokens, with 200 special tokens added for a maximum budget of 10000 tokens.
- **Fixed Interval (250 tokens):** Control tokens are inserted at a larger, fixed interval of every 250 tokens, with 40 special tokens added for a maximum budget of 10000 tokens.
- **Budget Ratio (Default):** Our proposed method, where 8 tokens are inserted at intervals corresponding to each 1/8th of the total budget.

Note that while the "No Control Tokens" and "Budget Ratio" strategies are inherently scalable to any context length, fixed-interval strategies require adjustments to the number of special tokens for different budget sizes.

The results, presented in Figure 3a, indicate that the **Budget Ratio** strategy yields the best performance. This suggests that a relative, ratio-based notification of remaining resources is more intuitive for the LLM to learn and adapt its reasoning pace, compared to fixed-interval notifications. Furthermore, the 250-token interval outperforms the 50-token interval, implying that a sparser, less intrusive set of control signals is more beneficial. The superior performance of our default strategy over the "No Control Tokens" baseline further validates the effectiveness of our core design.

| Model | Accuracy (%) | | | | Budget Following Ratio (%) | | | |
|---|---|---|---|---|---|---|---|---|
| | $2K$ | $4K$ | $6K$ | $10K$ | $2K$ | $4K$ | $6K$ | $10K$ |
| BudgetThinker 6k | 79.6 | 85.2 | 86.0 | 84.6 | 47.0 | 67.8 | 76.2 | 85.6 |
| BudgetThinker 6k-4k | 80.0 | 86.0 | 84.6 | 86.4 | 62.8 | 78.6 | 85.8 | 93.8 |
| BudgetThinker 6k-4k-3k | 80.0 | 84.2 | 85.0 | 87.2 | 66.4 | 84.6 | 89.8 | 95.4 |
| BudgetThinker 6k-4k-3k-2k | 83.6 | 84.6 | 84.6 | 85.2 | 86.8 | 95.4 | 97.2 | 98.6 |
| BudgetThinker (Full Training) | 81.6 | 84.8 | 85.8 | 87.4 | 79.8 | 95.4 | 95.0 | 97.2 |

Table 2: This table tracks the Accuracy (%) and Budget Following Ratio (%) of BudgetThinker-1.5B at different stages of the iterative training. Each row represents a checkpoint (*e.g.*, "6k-4k" denotes the model after training sequentially on 6k and 4k budgets). The "Full Training" model completes the 6k-to-2k curriculum plus a final mixed-budget training phase where budgets are randomly sampled. Darker cells indicate higher values within each column.

## 3.4 ANALYSIS OF ITERATIVE TRAINING

To investigate the impact of our mixed budget reinforcement learning, we evaluate checkpoints of BudgetThinker-1.5B on MATH-500, as shown in Table 2. The results reveal that as the model is trained on smaller budgets, its reasoning capability on larger budgets initially decreases. However, after the final mixed-budget training stage, accuracy on larger budgets recovers. We also observed that training on smaller budgets improves the budget following ratio, encouraging the model to generate more concise answers. After the full curriculum, the LLM achieves a balanced capability to handle all budgets, even if it is not individually optimal for every single budget constraint.

## 3.5 ABLATION ON REINFORCEMENT LEARNING

To isolate the contribution of reinforcement learning, we compare the full BudgetThinker model with a version trained only with Supervised Fine-Tuning (SFT). The results in Figure 3b demonstrate that the full RL-tuned model consistently surpasses the SFT-only model across all tested budgets. This underscores the importance of the RL phase. While SFT teaches the model the format and semantic meaning of control tokens, RL encourages the model to actively explore and discover more effective reasoning strategies that optimize the reward under specific budget constraints, ultimately leading to higher accuracy.

## 4 RELATED WORK

### 4.1 TEST-TIME SCALING FOR REASONING LLMS

Recent advances in test-time scaling seek to improve LLM reasoning by increasing computational depth during decoding. Among these test-time-scaling methods, reinforcement learning encourages LLMs to explore different strategies of solving problems and allocate more reasoning times for reflection. It has shown great potential in boosting LLM reasoning in multiple domains, including math (DeepSeek-AI, 2025; Google, 2025), coding (Hui et al., 2024; Yang et al., 2025), agentic tasks (Team, 2025a), and multimodal reasoning (Tan et al., 2025; Shen et al., 2025; Team et al., 2025b;a). Some other works propose to distill reasoning ability from long reasoning CoTs generated by large models to smaller LLMs to encourage deep thinking (Muennighoff et al., 2025; Labs, 2025; Team, 2025b; Ye et al., 2025; Xu et al., 2025a; Geiping et al., 2025). While effective at enhancing complex problem-solving, these methods often suffer from significant inference latency due to the generation of lengthy outputs (Sun et al., 2025; Zhu & Li, 2025; Qu et al., 2025), rendering them impractical for deployment in real-time systems (Wen et al., 2023; Jiang et al., 2021; Han et al., 2021). Besides, recent works also found that lengthy outputs often result in overthinking (*e.g.* meaningless repeats) and even lead to errors (Yu et al., 2025a; Chen et al., 2025b). Our method teaches LLMs how to follow a user-specified token budget during its reasoning process. This helps the model generate more controlled and efficient reasoning.

## 4.2 EFFICIENT CHAIN OF THOUGHTS

Several methods have been proposed to alleviate the token efficiency problem of LLMs' reasoning. Prompt methods (Xu et al., 2025b; Han et al., 2024; Muennighoff et al., 2025) make LLMs generate less reasoning tokens directly by adding explicit length constraints into the prompts. Solution routing works (Yu et al., 2025c; Wang et al., 2025a) allow for mid-generation control to prune unpromising traces. Computation routing methods (Damani et al., 2024; Snell et al., 2025; Fu et al., 2025a; Wang et al., 2025b; Fu et al., 2025b; Li et al., 2025; Yu et al., 2025b) choose to allocate just the necessary computation resources for solving problems, such as routing to larger models or conducting more samplings to vote for the final answer. There are also learning methods that either construct well-designed datasets to apply model fine-tuning (Chen et al., 2025b; Han et al., 2024; Zeng et al., 2025; Xia et al., 2025; Kang et al., 2025; Munkhbat et al., 2025; Liu et al., 2024) or use Reinforcement Learning (RL) methods (Yu et al., 2025a; Yang et al., 2025; Aggarwal & Welleck, 2025; Hou et al., 2025; Arora & Zanette, 2025) to encourage model to generate concise yet accurate answers. More recently, models like Qwen3 (Yang et al., 2025), Kimi k1.5 (Team et al., 2025c) and GPT-5 (OpenAI, 2025a) integrate hybrid thinking modes of long and short CoTs, seeking trade-off between output length and model performance. Instead, our work achieves precise token budget control. LLMs are trained to understand and follow the constraint in prompts and adaptively adjust their reasoning process according to the inserted special control tokens to complete the task within the user-requested budgets, showcasing the flexibility and efficiency of our method.

## 4.3 CoT CONTROLLING BY TOKEN INSERTION

Inserting tokens during generation to control LLM behavior has been explored for various objectives. In safety alignment, intervention sequences are strategically inserted at trigger points (*e.g.* , unsafe words) to steer the model towards safer outputs without retraining (Jain et al., 2025; Wu et al., 2025a; Fei et al., 2025). Other works use inserted tokens to shape the reasoning process itself, such as injecting "Time's Up! Therefore the final answer is:" to conclude reasoning (Sun et al., 2025), adding "planning tokens" to encourage structured thought (Wang et al., 2024) or "pause tokens" to allocate more computation at points of uncertainty (Kim et al., 2025). These methods mostly are focused on the quality of reasoning, whereas our goal is to control its length. The works most similar to ours are those that use in-generation hints for length control. ThoughtTerminator (Pu et al., 2025) and ConciseHint (Tang et al., 2025) both operate by periodically injecting multi-token textual hints (e.g., "100 tokens remaining. I'll be back.") to encourage brevity or budget adherence. However, our approach employs *single special tokens* instead of multi-token textual hints, which is highly token-efficient and practical for small budgets where verbose hints are infeasible. Besides, unlike hint-based methods that rely on zero-shot instruction following, BudgetThinker is a *trainable framework*, which learns to treat them as precise control signals for budget adherence, enabling a level of fine-grained length control that hint-based methods cannot achieve.

## 5 CONCLUSION

In this work, we introduce BudgetThinker, a novel framework that manages the trade-off between reasoning quality and computational cost in LLMs. By using special control tokens and a two-stage training pipeline, BudgetThinker enables precise, budget-aware control over the model's reasoning length. Our experiments validate that this method achieves superior budget adherence while maintaining high accuracy on challenging benchmarks. Thus, BudgetThinker represents a significant step towards developing more efficient LLMs suitable for real-time, resource-constrained applications.

## 6 REPRODUCIBILITY STATEMENT

To ensure reproducibility, we provide detailed descriptions of our model architecture, training pipeline, and evaluation protocol in Sections 2 and 3. The implementation of our special control token mechanism, SFT dataset curation, and GRPO-based reinforcement learning are fully documented in the main text, with additional details such as dataset composition, training hyperparameters, and compressor prompts included in Appendix A. All experiments are repeated across multiple model scales (1.5B and 7B LLMs, 7B MLLMs) and computational budgets to confirm stability of results. We additionally perform ablation studies on token insertion strategies, curriculum training, and RL contribution to validate robustness. The full codebase, processed datasets, and evaluation scripts will be released to the research community to facilitate replication and further exploration of budget-aware reasoning.

## 7 ETHICS STATEMENT

Our work investigates methods for improving the efficiency of LLM and MLLM reasoning under explicit computational budgets. All experiments are conducted using publicly available datasets such as MATH-500, AMC 2023, ERQA, and SAT benchmarks, or synthetic data generated by teacher models. No private or sensitive data was used. Since BudgetThinker focuses on controlling reasoning length rather than altering semantic content, the adversarial risk of harmful generation is minimal. Nevertheless, we avoid releasing prompts or training data that could be misused to bias models toward unsafe behaviors. Our contributions aim to advance responsible, resource-aware AI research, making LLMs more suitable for latency- and cost-sensitive applications. We comply with licensing restrictions of all datasets and models used, and our data handling adheres to privacy and legal standards. The authors declare no conflicts of interest or external funding sources that could improperly influence this work. We welcome community feedback on any additional ethical considerations.

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

# A APPENDIX

## A.1 LLM USAGE STATEMENT

Large language models (*e.g.*, ChatGPT) were used only for minor language polishing and grammar correction. All ideas, experimental design, data analysis, and writing were conducted by the authors.

## A.2 LLM DATASET DETAILS

For SFT, we select all 1k reasoning samples from s1k (Muennighoff et al., 2025), 0.8k reasoning samples from LIMO (Ye et al., 2025), and 17k reasoning samples from Bespoke-Stratos-17k (Labs, 2025), and a subset from NuminaMath(LI et al., 2024), Math(Hendrycks et al., 2021). Due to computational resource constraints, we remove all question-answer pairs with Chain-of-Thought (CoT) lengths over 10,000 tokens.

For reinforcement learning, the training data is a set of 14k math problems, comprising two parts: one part from the *math* category of the prm800k dataset (Lightman et al., 2023), and the other part from the *numina* and *amc* labeled data in the PRIME-RL/Eurus-2-RL-Data(Cui et al., 2025) dataset.

## A.3 COMPRESSOR PROMPT

---
**Compressor Prompt Example**

You are an agent tasked with compressing a reasoning process into a shorter, clearer version for a robot. Given the task description, the reasoning process, and the final answer, your goal is to shorten the reasoning process while preserving as many important details and as much logical completeness as possible.

**Task:** `<image><image>` The images are frames from a video. The video shows a ... Carefully analyze the multiple-choice question above and reason through it step by step.

**Reasoning process:** In the initial frame, the red fire box is centered, ... Therefore, the camera was moving left during the video. `<answer>A</answer>`

**Note that:**
- You must keep the original tags `<think>`, `</think>`, `<answer>`, `</answer>` exactly as they are. Do not shorten, rename, or remove them.
- You cannot change the final answer. Only compress the reasoning process.
- The reasoning process should be as concise as possible while preserving logical flow and essential details.
- Return only the compressed reasoning process. Do not include any extra comments, explanations, or text.

Your answer:`<think>`

---

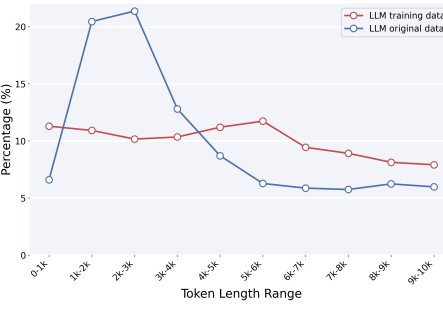 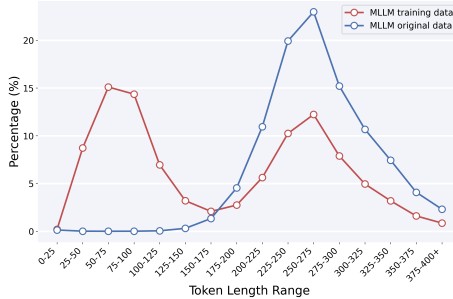

(a) Distribution between training and original data for LLM.

(b) Distribution between training and original data for MLLM.

Figure 4: Data distribution between the LLM and MLLM SFT datasets before and after augmentation.

## A.4 SFT DATASET DISTRIBUTION

To ensure the model generalizes across various reasoning lengths, we implemented a data augmentation strategy using GPT-4o-mini to generate compressed chain-of-thought variations. For the MLLM dataset (originally 37k), we produced two compressed versions (a first-round and a second-round compression), expanding the dataset to 111k. For the LLM dataset (originally 21k), we generated one compressed version, bringing the total to 41k.

Figure 4b compares the length distributions of the original data against the augmented training set. As illustrated, the original data was heavily concentrated in specific length ranges. The inclusion of compressed data successfully expands the distribution to cover a much broader spectrum of answer lengths.

## A.5 TRAINING AND EVALUATION DETAILS

The granularity parameter for generating training budgets is set to $T = 50$. The importance hyper-parameters are set to $k_1 = 0.7, k_2 = 0.15, k_3 = 0.15$. The length reward penalty coefficient $\gamma$ is set as 1 and 16 when $|y| \leq B$ and $|y| > B$ respectively.

During evaluation, we generate $N$ candidate solutions using sampling with a temperature t and a top-p value of k. We report the pass@1 accuracy, with each question sampled $N$ times and calculate the averaged accuracy. Following DeepSeek-R1 (DeepSeek-AI, 2025), we set $N$ depending on the size of test dataset, where $N = 1$ for MATH-500 ($t = 0, k = 1$) and $N = 64$ for AMC 2023 (for the 7B model, due to computational resource constraints, $N = 8$).

## A.6 LATENCY OVERHEAD

The following Table 3 compares the latency overhead differences between our method and the c2f, Fixed Intervals, and Thought Terminator (Pu et al., 2025) approaches. It shows that our method uses significantly fewer additional tokens compared to Fixed Intervals and Thought Terminator, while being particularly close to c2f.

| Model | Latency Overhead (tokens) | | | | | |
|---|---|---|---|---|---|---|
| | $B = 0.5K$ | $B = 1K$ | $B = 2K$ | $B = 4K$ | $B = 6K$ | $B = 10K$ |
| c2f | 7 | 7 | 7 | 7 | 7 | 7 |
| Fixed Intervals (250 Tokens) | 9 | 11 | 15 | 23 | 31 | 47 |
| Thought Terminator | 51 | 99 | 195 | 387 | 579 | 963 |
| BudgetThinker | 14 | 14 | 14 | 14 | 14 | 14 |

Table 3: Latency overhead of different methods under multiple budget settings.

**Question-Math1:**

Return your final response within \boxed{}. How many ways are there to put 5 balls in 2 boxes if the balls are not distinguishable and neither are the boxes? \n(Complete thinking within ███ tokens or fewer)<think>

*The following shows BudgetThinker's responses to the same problem under different budgets.*

**BudgetThinker (budget=500):**

<think>\nTo determine the number of ways to put 5 indistinguishable balls into 2 indistinguishable boxes, we need to consider the different partitions of the number 5 into at most 2 parts. Each part represents the number of balls in a box, and since the boxes are indistinguishable\n<remaining>7/8</remaining>\n, the order of the parts does not matter.\n\n1. **Partitions of 5 into at most 2 parts:**\n   - The possible partitions are:\n     - \\(5 = 5 + 0\\)\n     - \\(5 = 4 + 1\\)\n     - \\(5 = \n<remaining>6/8</remaining>\n3 + 2\\)\n\n2. **Verification of partitions:**\n   - \\(5 + 0\\) corresponds to one box containing all 5 balls and the other box empty.\n   - \\(4 + 1\\) corresponds to one box containing 4 balls and the other containing 1 ball.\n   -\n<remaining>5/8</remaining>\n \\(3 + 2\\) corresponds to one box containing 3 balls and the other containing 2 balls.\n\n3. **Counting the partitions:**\n   - There are three distinct partitions of 5 into at most 2 parts, each corresponding to a unique way of distributing the balls into the boxes.\n\nThus\n<remaining>4/8</remaining>\n, the number of ways to put 5 indistinguishable balls into 2 indistinguishable boxes is given by the number of partitions of 5 into at most 2 parts, which is 3.\n\n\\[\n\\boxed{3}\n\\]\n</think>\n<remaining>3/8</remaining>\n\n<remaining>2/8</remaining>\n\n\n<remaining>1/8</remaining>\n\n</think>\n**Final Answer**\n\\boxed{3} ✅

**BudgetThinker (budget=1000):**

<think>\n\nOkay, let's see. The problem is asking how many ways there are to put 5 balls into 2 boxes where the balls and the boxes are not distinguishable. Hmm, so both the balls and the boxes are identical. I need to figure out the different ways to distribute these identical balls into identical boxes. \n\nFirst, since both the balls and boxes are indistinct, the order doesn't matter. That means if I put a certain number of balls in each box, it's the same as putting them in the other box because the boxes are indistinct. But wait, no, wait\n<remaining>7/8</remaining>\n. Wait, if the boxes are indistinct, then switching the boxes doesn't create a new arrangement. So, the problem is similar to partitioning the number 5 (the number of balls) into at most 2 parts, where each part represents the number of balls in each box. But since the boxes can be empty or not, the number of integer partitions of 5 into at most 2 parts, where the order doesn't matter. So for example, putting 3 balls in one box and 2 in the other is the same as putting 2 and 3\n<remaining>6/8</remaining>\n. So the partitions would be (5), (4,1), (3,2), (3,1,1), (2,2,1), (1,1,1,1,1). Wait, no, but since we are only putting into at most 2 boxes, we can't have more than 2 parts. So the partitions of 5 into at most 2 parts. Let's list them:\n\n1. 5\n2. 4 + 1\n3. 3 + 2\n<remaining>5/8</remaining>\n, so we can't include that\n5. 2 + 2 + 1 (three parts)\n\n6. 1 + 1 + 1 + 1 + 1 (five parts)\n\n\nSo actually, the valid partitions into at most 2 parts are 5, 4+1, 3+2. Wait, but 3+2 is two parts, and 5 is single part. Then, maybe that's the answer? So three ways. But wait, let me check again. Because sometimes when we have indistinct boxes, we might\n<remaining>4/8</remaining>\n have different possibilities.\n\nAlternatively, maybe another approach. Let me think. Since both the balls and the boxes are indistinct, the problem reduces to finding the number of equivalence classes under permutation of boxes. So for each possible distribution (a, b) where a + b = 5, and a, b are non-negative integers. But since the boxes are indistinct, (a, b) is the same as (b, a). Therefore, the number of distinct partitions is equal to the number of unordered pairs (a, b) where a + b = 5. So how many such pairs are there\n<remaining>3/8</remaining>\n?\n\nStarting from a = 0, b =5; a=1, b=4; a=2, b=3; then a=3, b=2; a=4, b=1; a=5, b=0. But since the boxes are indistinct, (0,5) is the same as (5,0), which is one case. Similarly, (1,4) and (4,1) are the same. (2,3) and (3,2) are the same. So how many unique cases? For 5, since 5\n<remaining>2/8</remaining>\n is odd, the middle case is (2,3). The rest are symmetric. So we have three distinct cases: (5), (4,1), (3,2). Therefore, the number of ways is 3.\n\nSo the answer should be 3.\n\n<remaining>1/8</remaining>\n\n</think>\n**Final Answer**\n\\boxed{3} ✅

**BudgetThinker (budget=2000):**

<think>\n\nOkay, let's try to figure out how many ways there are to put 5 indistinct balls into 2 indistinct boxes. Hmm, both the balls and the boxes are not labeled, so the order of the balls and the order of the boxes don't matter. That should simplify things a bit, right?\n\nFirst, when dealing with indistinct objects and boxes, this is a classic combinatorics problem. Normally, if the boxes were distinguishable, the problem would be about finding the number of non-negative integer solutions to the equation x + y = 5, where x is the number of balls in the first box and y is the number in the second. Since there are two variables, the number of solutions would be 6 (since x can be 0 to 5, and y would adjust accordingly). But in this case, the boxes are indistinct, so some of those solutions might be considered the same. Let me think.\n\nSo instead of 6, maybe it's half of that, but I have to be careful because in cases where x = y, there's no pair to pair with. Since 5 is an odd number, x can't equal y in this case. Because if they were equal, then x + y = 2.5, but since we have half a ball, that's impossible. So all solutions have x ≠ y. Therefore, the total number of unordered pairs would be the number of ordered pairs divided by 2.\n\nSince there are 6 ordered pairs, that would be 6 / 2 = 3. So maybe there are 3 ways?\n\nWait, let me check that again. Let's list all possible ordered pairs first: For distinguishable boxes, the ordered pairs (x, y) where x + y = 5 are:\n\n(0,5), (1,4), (2,3), (3,2), (4,1), (5,0). There are 6 of these. Since the boxes are indistinct, each pair where x ≠ y is counted once, and the pairs where x = y are counted once.\n\nBut\n<remaining>6/8</remaining>\n as x = y would require 2x = 5, which isn't possible here, so all 6 pairs are of the form (a, b) where a ≠ b. Each unordered pair is represented twice in the ordered list. Therefore, the number of unordered pairs is 6 / 2 = 3. So there are 3 distinct ways. Let me list them:\n\n1. Both boxes have different numbers of balls: (0,5) and (5,0) are the same.\n\n2. (1,4) and (4,1) are the same.\n\n3. (2,3) and (3,2) are the same.\n\nSo that's three unordered pairs. So, the answer should be 3. But wait, the answer should be 3. But wait, let me think again if that's really what anything\n<remaining>5/8</remaining>\n. Alternatively, maybe there's a different approach. Since there's no pair to pair with, but since they are indistinct, the problem reduces to considering the partitions of the number 5 into at most 2 parts. Because each box can hold any number of balls, but since they're indistinct, we can consider the number of distinct ways to partition 5 into a sum of two non-negative integers where order doesn't matter.\n\nThe partitions of 5 into at most 2 parts are:\n\n1. 5 + 0\n\n2. 4 + 1\n\n3. 3 + 2\n\nThese are the three partitions. Each of these corresponds to the unordered pairs (5,0), (4,1), and (3,2). So that's three ways. So that's why there are 3 ways. That confirms the previous result.\n\nAlternatively, if I remember the formula for the number of ways to divide n indistinct objects into k indistinct boxes, it's the floor of (n + k - 1 divided by k, adjusted for symmetry. Wait, no, that might be for something else. Let me think. Actually, when objects are indistinct and boxes are indistinct, the number of ways is equal to the number of integer partitions of n into at most k parts. For\n<remaining>4/8</remaining>\n k=2, it's the number of partitions of 5 into at most 2 parts.\n\nBut let me recall the partitions of 5. \n\nThe number of partitions of 5 into at most 2 parts is just: 5\n\n5\n\n4 + 1\n\n3 + 2\n\nSo that's 3. So that's the same as the previous result.\n\nWait, so total of 1 + 2 = 3. So that's the same as the previous cases. \n\nIn other words, the answer is 3. Then we have to pick those partitions that have 1 part: just 5. The partitions into 1 part: just 5.\n\n3+2. So total of 1 + 2 = 3. So that's the same as the previous result.\n\nWait, so there are 3 ways. Hmm, let me make sure I didn't miss any other partitions. For example, 3 + 1 + 1 would be 3 parts, so if k=2, we can't consider that. So indeed, only 3 ways.\n\nAlternatively, maybe someone could think that since the problem is about distributing balls into boxes, the answer is the combination formula. But since the objects are indist\n<remaining>3/8</remaining>\n\\ndinct and the boxes are indistinct, perhaps we need a different approach. For example, the formula for identical objects into identical boxes. The general formula for the number of ways is the number of integer partitions of n into at most k parts. So here, n=5, k=2, so number of partitions of 5 into at most 2 parts. As we saw, that's 3. So that's correct.\n\nTherefore, the answer is 3.\n<remaining>2/8</remaining>\n\n<remaining>1/8</remaining>\n\n</think>\n**Final Answer**\n\\boxed{3} ✅

**BudgetThinker (budget=4000):**

<think>\n\nOkay, let's try to figure out how many ways there are to put 5 indistinct balls into 2 indistinct boxes. Hmm, both the balls and the boxes are not labeled, so the order of the balls and the order of the boxes don't matter. That should simplify things a bit, right?\n\nFirst, when dealing with indistinct objects and boxes, this is a classic combinatorics problem. Normally, if the boxes were distinguishable, the problem would be about finding the number of non-negative integer solutions to the equation x + y = 5, where x is the number of balls in the first box and y is the number in the second. Since there are two variables, the number of solutions would be 6 (since x can be 0,1,2,3,4,5). But in this case, the boxes are indistinct, so some of these solutions can be grouped together. Like, having 0\n<remaining>7/8</remaining>\n on one box and 5 in the other is the same as having 5 in the first and 0 in the second. So, we need to adjust for that symmetry.\n\nIn the distinguishable box case, the number of ways is equal to the number of integer partitions of 5 into at most 2 parts. Since boxes are indistinct, the order doesn't matter. So, partitions of 5 into at most 2 parts. Let me list them:\n\n1. 5: 5 + 0: Since order doesn't matter, 5 + 0 is the same as 0 + 5.\n\nit's just one way.\n\n2. 4 + 1: Similarly, this is another distinct partition.\n\n3. 3 + 2: That's the third one.\n\nAre there more? Let's see. 5 can be partitioned as\n<remaining>6/8</remaining>\n 5\n= 4 + 1\n= 3 + 2\n= 2 + 2 + 1, but wait, that's three parts, and we're only allowed two boxes. So we can't have three parts. So only up to two parts.\n\nSo those are 3 partitions here. Therefore, there are 3 ways. But wait, let me verify this.\n\nAlternatively, maybe using the formula for combinations in indistinct objects\n<remaining>5/8</remaining>\n distinct boxes and indistinct balls. Wait, but the boxes are indistinct, so we count the number of distinct partitions. The formula for distributing n indistinct objects into\n<remaining>4/8</remaining>\n indistinct boxes is C(n + k - 1, k - 1). So here, n = 5, k = 2, so it would be C(5 + 2 - 1, 2 - 1) = C(6,1) = 6. So there are 6 ways when the boxes are indistinct, then the different distributions where the number of balls in the boxes are considered the same. So we need to count the number of distinct partitions. Since each partition corresponds to a unique way of distributing pairs (a, b) where a + b = 5, and a ≤ b (to avoid duplicates). Because if a + b = 5, then swapping them would give the same distribution, so we count each pair only once. When a = b, we don't need to swap.\n\n<remaining>3/8</remaining>\nThat's three pairs. So these are 3 distinct ways. That matches the partition idea.\n\nWait, let me check if there are any other possibilities. For example, what if there's a case where a = b? Then 5 would be split. So there's no way where a = b here. So that's not possible. So all the partitions are \n<remaining>2/8</remaining>\n as above.\n\nTherefore, the answer should be 3. But to make sure I haven't missed anything, let me enumerate all possible distributions for the indistinct boxes. So the possible distributions (x, y) where x + y = 5 are\n<remaining>1/8</remaining>\n(0,5), (1,4), (2,3), (3,2), (4,1), (5,0). But since these are indistinct, (0,5) is the same as (5,0). So grouping them, we have three distinct cases: 0 and 5, 1 and 4, 2 and 3. So yes, three distinct ways. So the number of unordered pairs is 3. So if they are indistinct, this is the same again to make sure. Maybe another approach: using combinations.\n\nIf the boxes are indistinct, the number is 6. If they are distinct, then the formula for combinations would be C(n + k - 1, k - 1). Here, n = 5, k = 2, so C(6,1) = 6. All in one box: 5 in one, 0 in other. Since they are indistinct, this is another way.\n<remaining>2/8</remaining>\n\n<remaining>1/8</remaining>\n, and 1 + 4 in one, 2 + 3 in the other. So we don't count it again.\n\nHence, three ways. I can't think of any other distribution. For example, 0 and 5, 1 and 4, 2 and 3. So we don't need 5 in one, 0 in other. So yes, three ways.\n\nTherefore, the final answer is 3, which would be written as\n<remaining>1/8</remaining>\n\n</think>\n**Final Answer**\n\\boxed{3} ✅

**BudgetThinker (budget=6000):**

<think>\n\nOkay, let's try to figure out how many ways there are to put 5 indistinct balls into 2 indistinct boxes. Hmm, both the balls and the boxes are not labeled, so the order of the balls and the order of the boxes don't matter. That should simplify things a bit, right?\n\nFirst, when dealing with indistinct objects and boxes, this is a classic combinatorics problem. Normally, if the boxes were distinguishable, the problem would be about finding the number of non-negative integer solutions to the equation x + y = 5, where x is the number of balls in the first box and y is the number in the second. Since there are two variables, the number of solutions would be 6 (since x can be 0,1,2,3,4,5). In the distinguishable box case, the number of ways is equal to the number of integer partitions of 5 into at most 2 parts. Since boxes are indistinct, the order doesn't matter. So, partitions of 5 into at most 2 parts. Let me\n<remaining>7/8</remaining>\n list them:\n\n5 + 0: But since the boxes are indistinct, the order doesn't matter. So, if we think of the distributions as unordered pairs (x, y), then the distinct partitions would be the same. 5 + 0 is the same as 0 + 5. So that's one way.\n\n4 + 1: Similarly, 1 + 4 and 4 + 1. But since they're indistinct, the order doesn't matter. So that's another way.\n\n3 + 2: Similarly, 2 + 3. So, the distinct partitions would be (5), (4,1), (3,2). So, that's three\n<remaining>6/8</remaining>\n partitions. Therefore, there are 3 ways to put 5 indistinct balls into 2 indistinct boxes.

Wait, but let me check this all out. Let me list all possible partitions of 5 into at most 2 parts: 5, 4+1, 3+2. Three partitions. Therefore, the answer is 3. But wait, I want to make sure I'm not missing any other partitions. For example, 5 = 5, 5 = 4 + 1, 5 = 3 + 2, and if we allow three or more parts, but since we have only 2 boxes, we can't have more than 2 parts.\n<remaining>5/8</remaining>\n So, 5 = 5, 4 + 1, 3 + 2. So, the distinct partitions would be (5), (4,1), (3,2). That's all of it. So, the answer is 3. But wait, I want to make sure I'm not missing any other partitions. For example, 5 = 5, 5 = 4 + 1, 5 = 3 + 2, and if we allow three or more parts, but since we have only 2 boxes, we can't have more than 2 parts. So, 5 = 5, 4 + 1, 3 + 2. So, the distinct partitions. Let's say we need to generate the functions. The generating function for indistinct objects into indistinct boxes would be something like 1 + x + x^2 + x^3 + x^4 + x^5 ... but since the boxes are indistinct and we can have any number of balls in each, the generating function for two boxes would be the coefficient of x^5 in the generating function (1 + x + x^2 + x^3 + x^4 + x^5)^2. Wait, but generating functions for partitions into at most 2 parts. Let me, me... actually, for two indistinct boxes, I think, actually, for two indistinct boxes, the generating function would be the generating function for partitions of 5 into at most 2 parts\n<remaining>4/8</remaining>\n when the number of objects is small, we can enumerate all possible distributions regardless of the formula. Let me just list them. We only have 2 balls, 2 boxes. What about (2,3)? That seems to make sure, let's think of all possible distributions when boxes are indistinct. So, if n = 5, and the other is 0. But the other one has 5 balls. So one box has all 5 balls, the other has 0. Then there is another unique way.\n<remaining>3/8</remaining>\n unique way.\n\nSo then there's one more: 3 and 2, which is another unique way.\n<remaining>2/8</remaining>\n. So, that's the other has 2. Similarly, 4 and 1 is another unique way.\n\nHence, three ways. So, the other has 3. Again, since they are indistinct, this is another unique way too. So there has 3, the other has 2. Again, since they are indistinct, this is another unique\n<remaining>1/8</remaining>\n unique way.\n\nSo\n<remaining>5/8</remaining>\n unique way.\n\nSo, only the three unique distributions.\n<remaining>4/8</remaining>\n 3 ways.\n<remaining>3/8</remaining>\n\n<remaining>2/8</remaining>\n\n<remaining>1/8</remaining>\n\n</think>\n**Final Answer**\n\\boxed{3} ✅

Figure 5: This is a comparison between BudgetThinker's responses to the Question-Math1 under different budgets.

## A.7 Detailed Analysis

For the qualitative analysis of the reasoning strategy variations of the BudgetThinker model under different budget constraints, Figure 5 presents multiple responses generated by the BudgetThinker model for the same problem, Question-Math1, across various budget allocations.

It can be observed that as the reasoning budget is systematically reduced from 6000 to 500, the thought process exhibits a progressive compression in both methodological diversity and explanatory depth, along with the omission of certain steps. For a detailed analysis of the reasoning process, refer to Figure 6.

At the highest budget of 6000, the reasoning is most comprehensive and expanded, covering multiple methods (e.g., ordered pair enumeration, unordered pair deduplication, integer partitions, symmetry

analysis, generating functions) and containing extensive metacognitive statements and pedagogical explanations. For instance, the generating function method is fully elaborated: *"Alternatively, maybe using generating functions. . . The generating function for indistinct objects. . . coefficient of $x^5$ in $(1 + x + \cdots + x^5)^2$"*, alongside dense self-verification loops such as *"Let me verify this."* and *"Wait, but let me think again."*.

When the budget is reduced to 4000, the reasoning shows multi-route, multi-round verification characteristics, but the depth and cyclicality of each method chain are compressed compared to 6000; method coverage remains broad (including ordered pairs, partitions, combinatorial formulas, test cases, etc.), but advanced methods like generating functions are only mentioned without elaboration, e.g., *"But wait, let me verify this."* and *"Suppose instead of 5 balls. . . 1 ball. . . the answer should be 1. . . "* demonstrate light reflection and the use of test cases.

Further compression occurs at a budget of 2000, reasoning retains multiple paths (e.g., full ordered pair enumeration, symmetric halving, partition verification) but each path is more linear, such as completely listing ordered pairs: *"x + y = 5. . . ordered pairs: (0,5), (1,4), (2,3), (3,2), (4,1), (5,0). There are 6."*, followed by symmetric grouping: *"Since all are of the form a ≠ b, unordered pairs = 6 / 2 = 3."*, yet lacks advanced methods (e.g., generating functions) and complex background explanations.

At a budget of 1000, reasoning is further compressed, retaining only core branching and slight hesitant language, e.g., *"But wait, no, wait. . . If the boxes are indistinct. . . the problem reduces to partitions. . . "*, and introduces a second method (unordered pairs) for one-time verification: *"(0,5),(1,4),(2,3). . . Since the boxes are indistinct. . . we have three distinct cases."*, but verification is limited and lacks redundant loops.

Finally, at the most constrained budget of 500, reasoning is extremely highly compressed, adopting a single linear path with highly structured language and no hesitation or alternative method discussion, for example, directly listing partitions: *"Partitions of 5 into at most 2 parts: 5 = 5 + 0, 4 + 1, 3 + 2"*, and performing minimal verification: *"Verification: 5+0 corresponds to . . . 4+1 corresponds to . . . 3+2 corresponds to. . . "*, with the entire logical path being linear and without branching.

Lastly, a comparative analysis of the responses generated by the BudgetThinker model against other models on both the VQA and mathematical reasoning tasks is presented in Figures 9, 7, and 8, accompanied by a detailed examination of BudgetThinker's advantages in handling diverse question types.

| Budget | Number of reasoning steps | Primary steps | Verification |
|---|---|---|---|
| 500 | 3 steps | 1. Reformulate the problem as integer partitions. 
 2. Enumerate the three partitions. 
 3. Briefly verify the semantic meaning of each partition and count them. | Minimal — a single brief verification (e.g., "one box containing 5… 4+1… 3+2…"). |
| 1000 | 7 steps | 1. Explain the meaning of "indistinguishable." 
 2. Temporarily list incorrect partitions (e.g., 3+1+1). 
 3. Exclude partitions with >2 parts. 
 4. Re-list the valid partitions. 
 5. Re-derive the result using unordered-pair reasoning. 
 6. Symmetry check (odd-case midpoint). 
 7. Conclude. | Moderate — at least two reflections / re-derivations (explicit "wait, no… let me check again" phenomenon). |
| 2000 | 10 steps | 1. Consider distinguishable boxes (solve x+y=5 → 6 nonnegative solutions). 
 2. Address indistinguishability and justify dividing by 2. 
 3. Enumerate ordered pairs. 
 4. Symmetry analysis (x = y impossible here). 
 5. Conclude unordered count = 3. 
 6. Return to partition-based explanation. 
 7. List all 7 partitions of 5. 
 8. Filter to "at most 2 parts" → 3. 
 9. Revisit combinatorial formula and reflect. 
 10. Multiple cross-checks. | Extensive verification — ≈ three repeated derivations + multiple-method cross-validation. |
| 4000 | 12 steps | 1. Explicitly contrast distinguishable vs. indistinguishable boxes ("boxes are indistinct… adjust for that symmetry"). 
 2. Apply stars-and-bars / combinations for the distinguishable case: $(\binom{5+2-1}{2-1}=\binom{6}{1}=6)$. 
 3. Motivate halving via symmetry / impose $(a\le b)$ (list $(0,5;1,4;2,3)$). 
 4. Enumerate ordered pairs and group them. 
 5. Argue $(a=b)$ impossible since (5) is odd. 
 6. Give partition perspective (5, 4+1, 3+2). 
 7. Cross-validate enumeration against informal formula. 
 8. Use the bound $(a\in\{0,\dots,\lfloor5/2\rfloor\})$ to enumerate quickly. 
 9. Provide small-(n) sanity checks (e.g., (n=1,2,3)). 
 10. Discuss the interpretation of zero parts (empty boxes). 
 11. Reconcile multiple approaches to the same conclusion. 
 12. State final result (=3). | Multiple cross-validations (combinatorial formula, unordered-pair reasoning, partition listing, explicit enumeration); at least two distinct methods mutually confirm the outcome. |
| 6000 | 13 steps | 1. Precisely set up distinguishable vs. indistinguishable framing. 
 2. Provide the distinguishable-case stars-and-bars / ordered solutions: $((0,5),(1,4),\dots)$ (6 total). 
 3. Explain why indistinguishability collapses ordered solutions to unordered equivalence classes. 
 4. List unordered solutions with $(a\le b)$: $((0,5),(1,4),(2,3))$. 
 5. Fully enumerate all partitions of 5 (all 7). 
 6. Explain the logical filter "partitions into $(\le2)$ parts." 
 7. Offer generating-function verification (coefficient of (x^5) in $((1+x+\cdots)^2)$). 
 8. Clarify the relationship between allowing zero parts and the standard definition of integer partitions (parts $(\ge1)$). 
 9. Reconcile stars-and-bars and partition perspectives (equivalences and differences). 
 10. Run multiple small-(n) sanity checks ((n=1,2,3,4)). 
 11. Discuss the effect of object distinguishability on counts (comparative implications). 
 12. Synthesize all method outcomes into a final aggregate. 
 13. Reiterate the conclusion ("therefore, the answer is 3") for emphasis. | Very extensive validation: diverse methods (enumeration, generating functions, algebraic equations, standard formulae), multiple test cases, and ≥3 independent confirmations. |

Figure 6: This is a detailed analysis of the reasoning processes generated by the BudgetThinker model for Question-Math1 under different reasoning budgets.

**Question-Math1:**
Return your final response within \boxed{}. How many ways are there to put 5 balls in 2 boxes if the balls are not distinguishable and neither are the boxes? \n(Complete thinking within `500/1000` tokens or fewer)<think>

**BudgetThinker (budget=500):**
<think>\nTo determine the number of ways to put 5 indistinguishable balls into 2 indistinguishable boxes, we need to consider the different partitions of the number 5 into at most 2 parts. Each part represents the number of balls in a box, and since the boxes are indistinguishable\n<remaining>7/8</remaining>\n, the order of the parts does not matter.\n\n1. **Partitions of 5 into at most 2 parts:**\n   - The possible partitions are:\n     - \\(5 = 5 + 0\\)\n     - \\(5 = 4 + 1\\)\n     - \\(5 = \n<remaining>6/8</remaining>\n3 + 2\\)\n\n2. **Verification of partitions:**\n   - \\(5 + 0\\) corresponds to one box containing all 5 balls and the other box empty.\n   - \\(4 + 1\\) corresponds to one box containing 4 balls and the other containing 1 ball.\n   - \n<remaining>5/8</remaining>\n \\(3 + 2\\) corresponds to one box containing 3 balls and the other containing 2 balls.\n\n3. **Counting the partitions:**\n   - There are three distinct partitions of 5 into at most 2 parts, each corresponding to a unique way of distributing the balls into the boxes.\n\nThus\n<remaining>4/8</remaining>\n, the number of ways to put 5 indistinguishable balls into 2 indistinguishable boxes is given by the number of partitions of 5 into at most 2 parts, which is 3.\n\n\\[\n\\boxed{3}\n\\]\n\n<remaining>3/8</remaining>\n\n<remaining>2/8</remaining>\n\n<remaining>1/8</remaining>\n\n</think>\n\n**Final Answer**\\boxed{3} ✅
**Total: 322 tokens**

**Original (budget=500):**
<think>\nOkay, so I have this problem: I need to figure out how many ways there are to put 5 balls into 2 boxes. The balls are not distinguishable, and neither are the boxes. Hmm, let me think about this step by step.\n\nFirst, since the balls are not labeled, it doesn't matter which specific ball goes into which box. Similarly, the boxes themselves aren't labeled, so swapping the contents of the boxes doesn't create a new arrangement. That simplifies things a bit because I don't have to worry about different box labels.\n\nLet me consider the possible distributions. Each ball has two choices: it can go into Box 1 or Box 2. Since there are 5 balls, each with 2 choices, the total number of ways without considering the boxes being indistinct would be 2^5, which is 32. But wait, that's if the boxes are distinguishable.\n\nSince they aren't, some of these 32 arrangements are duplicates.\n\nIf the boxes were distinguishable, like if one box was red and the other was blue, each ball independently choosing a box would give 2^5 = 32 unique arrangements. But since the boxes are the same, some of these are the same. For example, putting all 5 balls in Box 1 is the same as putting all 5 balls in Box 2 because the boxes are indistinct.\n\nSo, how many unique distributions are there? Well, each distribution can be characterized by how many balls are in Box 1 and how many are in Box 2. Since the total is 5, the number of balls in Box 1 can range from 0 to 5. But because the boxes are indistinct, putting k balls in Box 1 is the same as putting 5 - k balls in Box 2. So, for k = 0, it's the same as k = 5; for k = 1, it's the same as k = 4; and for k = 2, it's the same as k = 3. The middle case, k = 2.5, |CUT|✗
**Total: 500+10 tokens**

**BudgetThinker (budget=1000):**
<think>\n\nOkay, let's see. The problem is asking how many ways there are to put 5 balls into 2 boxes where the balls and the boxes are not distinguishable. Hmm, so both the balls and the boxes are identical. I need to figure out the different ways to distribute these identical balls into identical boxes. \n\nFirst, since both the balls and boxes are indistinct, the order doesn't matter. That means if I put a certain number of balls in each box, it's the same as putting them in the other box because the boxes are indistinct. But wait, no, wait\n<remaining>7/8</remaining>\n. Wait, if the boxes are indistinct, then switching the boxes doesn't create a new arrangement. So, the problem is similar to partitioning the number 5 (the number of balls) into at most 2 parts, where each part represents the number of balls in each box. But since the boxes can be empty or not, the problem is to find the number of integer partitions of 5 into at most 2 parts.\n\nInteger partitions where the order doesn't matter. So for example, putting 3 balls in one box and 2 in the other is the same as putting 2 and 3\n<remaining>6/8</remaining>\n. So the partitions would be (5), (4,1), (3,2), (3,1,1), (2,2,1), (1,1,1,1,1). Wait, no, but since we are only putting into at most 2 boxes, we can't have more than 2 parts. So the partitions of 5 into at most 2 parts. Let's list them:\n\n1. 5\n2. 4 + 1\n3. 3 + 2\n\n4. 3 + 1 + 1 (but that's three parts)\n<remaining>5/8</remaining>\n, so we can't include that)\n\n5. 2 + 2 + 1 (three parts)\n\n6. 1 + 1 + 1 + 1 + 1 (five parts)\n\nSo actually, the valid partitions into at most 2 parts are 5, 4+1, 3+2. Wait, but 3+2 is two parts, 4+1 is two parts, and 5 is single part. Then, maybe that's the answer? So three ways. But wait, let me check again. Because sometimes when we have indistinct boxes, we might\n<remaining>4/8</remaining>\n have different possibilities.\n\nAlternatively, maybe another approach. Let me think.\n\nSince both the balls and the boxes are indistinct, the problem reduces to finding the number of equivalence classes under permutation of boxes. So for each possible distribution (a, b) where a + b = 5, and a, b are non-negative integers. But since the boxes are indistinct, (a, b) is the same as (b, a). Therefore, the number of distinct partitions is equal to the number of unordered pairs (a, b) where a + b = 5. So how many such pairs are there\n<remaining>3/8</remaining>\n?\n\nStarting from a = 0, b =5; a=1, b=4; a=2, b=3; then a=3, b=2; a=4, b=1; a=5, b=0. But since the boxes are indistinct, (0,5) is the same as (5,0), which is one case. Similarly, (1,4) and (4,1) are the same. (2,3) and (3,2) are the same. So how many unique cases? For 5, since\n<remaining>2/8</remaining>\n is odd, the middle case is (2,3). The rest are symmetric. So we have three distinct cases: (5), (4,1), (3,2). Therefore, the number of ways is 3.\n\n\nSo, the answer should be 3.\n\n<remaining>1/8</remaining>\n\n</think>\n\n**Final Answer**\\boxed{3} ✅
**Total: 840 tokens**

**Original (budget=1000):**
<think>\n\nOkay, so I have this problem: I need to figure out how many ways there are to put 5 balls into 2 boxes. The balls are not distinguishable, and neither are the boxes. Hmm, let me think about this step by step.\n\nFirst, since the balls are not labeled, it doesn't matter which specific ball goes into which box. Similarly, the boxes themselves aren't labeled, so putting a ball in Box A is the same as putting it in Box B. That simplifies things a bit because I don't have to worry about which box is which; I just need to count the number of ways to distribute the balls.\n\nLet me consider the possible scenarios. Each ball has two choices: it can go into Box 1 or Box 2. Since there are 5 balls, each with 2 choices, I might think that the total number of ways is 2 multiplied by itself 5 times, which is 2^5. But wait, that would be 32. But hold on, that doesn't seem right because it counts all possible assignments, including the case where all balls are in one box and none in the other. But the problem doesn't specify that both boxes must have at least one ball, so maybe that's acceptable.\n\n\\n\nBut let me think again. If the boxes are indistinct, does that mean that putting all balls in Box 1 is the same as putting all in Box 2? So, in that case, the number of distinct distributions would be half of 2^5, which is 16. But wait, is that correct?\n\n\\n\nNo, actually, no. Because when you have two boxes, each ball independently goes into either box, so the total number of assignments is 2^5, but since the boxes are indistinct, some of these assignments are duplicates. For example, putting a ball in Box 1 and another in Box 2 is the same as a ball in Box 2 and another in Box 1. So, in reality, the number of distinct ways is actually the number of subsets of the 5 balls, because each subset represents the balls in Box 1, and the complement represents the balls in Box 2. Since the boxes are indistinct, the order doesn't matter, so the number of ways is equal to the number of subsets of a 5-element set, which is 2^5 divided by 2, but wait, that would be 16, but actually, the number of subsets is 2^5, which is 32, but since each subset and its complement are considered the same when boxes are indistinct, the number of distinct ways is 2^(5-1) = 16. Hmm, that seems conflicting.\n\nWait, maybe I'm overcomplicating it. Let me think about it differently. If the boxes are indistinct, then the number of ways to distribute the balls is equal to the number of ways to choose how many balls go into each box. Since the balls are indistinct, the only thing that matters is how many go into each box. So, for each possible number of balls in Box 1, from 0 to 5, the number of ways is 1 for each number, except when the number is the same as the complement, which is when the number is 2 or 3, because 2 and 3 are complements of each other in a 5-ball set.\n\nWait, no, that's not quite right. If I have 5 balls, the number of ways to choose k balls for Box 1 is C(5, k), where C is the combination function. But since the boxes are indistinct, choosing k balls for Box 1 is the same as choosing 5 - k balls for Box 2. So, the number of distinct distributions is equal to the number of distinct values of k, where k ranges from 0 to 5, but considering that k and 5 - k are the same when boxes are indistinct.\n\nSo, for k = 0, it's the same as k = 5. For k = 1, it's the same as k = 4. For k = 2, it's the same as k = 3. So, the distinct distributions are determined by the number of balls in Box 1, which can be 0, 1, 2, or 3. That's 4 distinct distributions. But wait, that doesn't seem right because 0 and 5 are the same, 1 and 4 are the same, and 2 and 3 are the same. So, actually, there are 3 distinct distributions: all balls in one box, 4 balls in one box and 1 in the other, or 2 and |CUT|✗
**Total: 1000+10 tokens**

**Comparison:**
Above are responses from two models—BudgetThinker and the Original model—to the same question under different budget constraints (500 and 1000).

It can be observed that under both budget conditions, BudgetThinker successfully completes its responses within the budget limits (322 and 840 tokens, respectively) while clearly following a compact, stepwise format and producing the final answer. In contrast, the Original model exhausts the allotted budget (500+ tokens and 1000+ tokens) by expanding irrelevant lines of thought and becomes truncated, failing to present a concise, correctly framed conclusion before budget exhaustion.

BudgetThinker: Our BudgetThinker reduces the problem immediately to integer partitions of 5 into at most 2 parts and lists the valid partitions ((5), (4+1), (3+2)). This direct partition viewpoint eliminates unnecessary combinatorial detours, verifies each case succinctly, and counts them to produce (\boxed{3}). The chain of thought is minimal, tightly scoped to the mathematical equivalence (distribution ↔ partition), and avoids redundancy — yielding a correct, compact answer within budget.

Original model: The Original model spends tokens on irrelevant expansions (treating each ball as independently choosing a box → (2^5), then trying to reconcile subset/complement reasoning, repeatedly rephrases the same ideas, and toggles between incompatible perspectives. This redundancy both wastes budget and creates internal contradictions; as a result the output is truncated and the final answer is not reliably delivered. The process lacks the direct, partition-based insight that would have yielded a short, correct result.

Figure 7: This is a comparison between BudgetThinker and Original model's response cases on MATH datasets.

**Question-Math2:**
Return your final response within \\boxed{}. What is the least positive integer multiple of 30 that can be written with only the digits 0 and 2? \n(Complete thinking within 500/2000 tokens or fewer)<think>

**BudgetThinker:**
<think>\n1. **Identify the properties of the number:**\n  - The number must be a multiple of 30.\n  - The number must be written using only the digits 0 and 2.\n\n2. **Check the divisibility rules:**\n  - A number is divisible by\n<remaining>7/8</remaining>\n 30 if and only if it is divisible by 2, 3, and 5.\n  - For a number to be divisible by 2, its last digit must be 0.\n  - For a number to be divisible by 5, its last digit must be 0 or 5\n<remaining>6/8</remaining>\n. Since we can only use the digits 0 and 2, the last digit must be 0.\n  - For a number to be divisible by 3, the sum of its digits must be divisible by 3.\n\n3. **Construct the number:**\n  - Since the number must end in \n<remaining>5/8</remaining>\n0, we start with the smallest possible number ending in 0.\n  - We need to find the smallest number composed of the digits 0 and 2 that is divisible by 3.\n\n4. **Check the sum of digits:**\n  - The sum of the digits of the number must be divisible by\n<remaining>4/8</remaining>\n 3.\n  - The smallest number composed of the digits 0 and 2 that satisfies this condition is 2220, where the sum of the digits is \\(2 + 2 + 2 + 0 = 6\\), which is divisible by 3.\n\n5. **Verify the\n<remaining>3/8</remaining>\n number:**\n  - The number 2220 ends in 0, so it is divisible by 2 and 5.\n  - The sum of its digits is 6, which is divisible by 3.\n  - Therefore, 2220 is divisible by 30.\n\nConclusion\n<remaining>2/8</remaining>\n:\n\\[\n\\[\n\\boxed{2220}\n\\]\n\\n\n<remaining>1/8</remaining>\n\n</think>\n**Final Answer**\n\\boxed{2220} ✅

`Total: 409 tokens`

**Original:**
<think>\nOkay, so I need to find the smallest positive integer that's a multiple of 30 and only uses the digits 0 and 2. Hmm, let's break this down. First, multiples of 30 are numbers that can be divided by 30 without any remainder. Since 30 is 2 times 3 times 5, the number must be divisible by 2, 3, and 5.\n\nDivisibility by 2 is straightforward: the number must end with an even digit. Since we can only use 0 and 2, that's already satisfied because both are even. \n\nNext, divisibility by 5 means the number must end with a 0 or 5. But since we can only use 0 and 2, it must end with a 0. So the number has to end with 0.\n\nNow, for divisibility by 3, the sum of all digits must be a multiple of 3. Since we can only use 0s and 2s, the sum will be 2 times the number of 2s in the number. So, the number of 2s must be a multiple of 3. That means the number must have 3, 6, 9, etc., 2s.\n\nSo, putting it all together, the number must end with a 0 and have a number of 2s that's a multiple of 3. Let's start with the smallest possible number.\n\nThe smallest number would be a 3-digit number because a 2-digit number can't have enough 2s to be divisible by 3. Let's see: 2220. Is that divisible by 30? Let's check:\n2220 divided by 30 is 74, which is an integer. So, 2220 is a multiple of 30. But wait, is there a smaller number? Let's check if a 3-digit number with fewer 2s can work. The next possible number of 2s is 3, so 2220 is the smallest.\n\nWait, hold on. 2220 is 222 followed by 0. But is there a smaller number? Let's think about 2220. Maybe a 4-digit number with fewer 2s? But 2220 is 4 digits, and if we try to make a|CUT|✖

`Total: 500+10 tokens`

---

**BudgetThinker:**
<think>\n\nOkay, so I need to find the smallest positive integer that's a multiple of 30 and only uses the digits 0 and 2. Let me start by understanding the problem. \n\nFirst, a multiple of 30 must be divisible by 30. Since 30 is 2*3*5, the number has to be divisible by 2, 3, and 5. Let's break that down:\n\n\n1. **Divisible by 2**: The number must end with an even digit. Since we can only use 0 or 2, the last digit has to be 0 or 2. But since 0 is even and 2 is even, both are okay. However, if the last digit is 0, the number is even, and if it's 2, it's also even. So that's fine.\n\n\n2. **Divisible by 5**: The number must end with 0 or 5. But since we can only use 0 and 2, the number must end with 0. So the number must end with 0. That simplifies things because the last digit is fixed as 0.\n\n\n3. **Div\n<remaining>7/8</remaining>\nisible by 3**: The sum of the digits must be a multiple of 3. Since the number is composed of 0s and 2s, the sum of digits is just 2 times the number of 2s. So, the number of 2s must be a multiple of 3/2. Wait, that might not make sense. Let me think again. If the sum of the digits is 2*k (where k is the number of 2s), then 2*k must be a multiple of 3 because 2 and 3 are coprime. So the number of 2s must be 3, 6, 9, etc.\n\nSo putting this together, the number must end with 0, have only 0s and 2s, and have a number of 2s that is a multiple of 3. Also, it needs to be the smallest such number.\n\nLet me start constructing numbers. The smallest possible numbers would be 20, 220, 2220, etc., but let's check each.\n\n\n- **20**: Ends with 0, digits are \n<remaining>6/8</remaining>\n0, digits are 2,0. Sum of digits is 2, which is not divisible by 3. So 20 is not divisible by 3, hence not by 30.\n\n\n- **220**: Ends with 0. Digits are 2,2,0. Sum is 4, not divisible by 3. Not good.\n\n\n- **2220**: Ends with 0. Digits are 2,2,2,0. Sum is 6, which is divisible by 3. Let's check if it's divisible by 2 and 5. Yes, it ends with 0. So 2220 is divisible by 2, 3, and 5. Therefore, 2220 is a multiple of 30.\n\nSo 2220 is the next candidate. Let's check if there's a smaller number with fewer digits.\n\nWait, the numbers we considered so far are 20, 220, 2220. Let's check if there's a 3-digit number with two 2s and one 0. The possible numbers are 220, 202\n<remaining>5/8</remaining>\n or 022. 022 is not a valid three-digit number.\n\n2, 022. But 022 is not a 3-digit number, it's a 2-digit number 22. So 220 is the only 3-digit number with two 2s and a 0. But 220 isn't divisible by 3. Then 2220 is the next candidate.\n\n\nBut maybe there's a 4-digit number with three 2s and a 0. For example, 2220. That's the one we have. Let's check if there's a smaller number with three 2s and a 0. The smallest such number would be 2220, right? Because if you arrange the 0 in the higher digits, like 2202, but that's 2202. Let's check 2202.\n\n\n- **2202**: Ends with 2, so not divisible by 5. So that's out.\n\n\n- **2022**: Ends with 2, same problem. Not divisible by 5.\n\n\n- **0222**: That's 222, which is a 3-digit number. But 2\n<remaining>4/8</remaining>\n22 is not divisible by 2, since it ends with 2, which is out with 2, which is even. Wait, 222 ends with 2, so it's not divisible by 5. So 222 is even, but not divisible by 3, so 222 is out. Then 2220 is the next candidate. Let's check if there's a 4-digit number with four 2s and a 0. But that would be larger than 2220, so 2220 is smaller.\n\n\nWait, but maybe there's a number with more digits but arranged in a way that the 0 is in a lower place, but the number is smaller. For example, 20220, but that's larger than 2220. So 2220 seems to be the smallest.\n\n\nBut let me confirm. Let's check all numbers with digits 0 and 2, ending with 0, and with sum of digits\n<remaining>3/8</remaining>\n divisible by 3.\n\nThe sum of digits is 2k, which must be divisible by 3. So k must be 3,6,9,... So the number must have 3,6,9,... number of 2s. The smallest such number would be with 3 2s, which is 2220. If we have 6 2s, that's a larger number, so 2220 is the smallest.\n\nTherefore, the answer should be 2220.\n\nTo find the least positive integer multiple of 30 using only the digits 0 and 2, we analyze the requirements for divisibility:\n\n1. **Divisibility by 2**: The number must end in 0 (since 2 is not allowed in digits).  3. **Divisibility by 3**: The sum of digits must be divisible by 3.  Since the number can only contain 0s and 2s: the sum of digits is \\(2k\\), where \\(k\\) is the number of 2s. \n\\(2\n<remaining>2/8</remaining>\nk\\) must be divisible by\\(k\\) must be a multiple of 3.  The smallest valid \\(k\\) is 3 (since \\(k=0\\) results in a single-digit number 0, which is trivial).  Constructing the smallest number with 3 twos and ending in 0:  - The smallest arrangement is \\(2220\\).  **Verification**:  - **Divisibility by 2**: Ends in 0.  - **Divisibility by 5**: Ends in 0.  - **Divisibility by 3**: Sum of digits \\(2 + 2 + 2 + 0 = 6\\), which is divisible by 3.  Thus, \\(2220\\) is divisible by 30. Smaller numbers (e.g., 20, 220, 222) fail \n<remaining>1/8</remaining>\n divisibility by 3 or 5.\n\n\n</think>\n**Final Answer**\\boxed{2220} ✅

`Total: 1723 tokens`

**Original:**
<think>\nOkay, so I have this math problem here: I need to figure out how many different values I can get from the expression 2·3·4·5 + 1 by inserting parentheses. The note says that rearranging terms isn't allowed, only inserting parentheses. Hmm, interesting. Let me try to break this down step by step.\n\nI know that without any parentheses, the expression is evaluated as 2 multiplied by 3, then multiplied by 4, then multiplied by 5, and finally add 1. Let me calculate that real quick: 2x3 is 6, then 6x4 is 24, 24x5 is 120, and 120 + 1 is 121. So, the original expression equals 121.\n\nBut the question is about inserting parentheses to change the order of operations and get different results. So, I need to explore all possible ways to insert parentheses in the expression 2·3·4·5 + 1 and see what values I can get.\n\nLet me write down the expression with the numbers and the operations: 2·3·4·5 + 1. The operations here are multiplication and addition. Since multiplication has higher precedence than addition, without any parentheses, it's evaluated as 2x3x4x5 + 1.\n\nNow, inserting parentheses can change the order in which operations are performed. Since we have two operations: multiplication and addition, parentheses can group either the multiplication part or the addition part. Since addition is only between 5 and 1, I can't really group the addition part unless I group the entire expression as a single term, which doesn't make sense because it's 2·3·4·5 + 1.\n\nWait, actually, the expression is 2·3·4·5 + 1, so the addition is already there.\n\nSo, maybe the only way to change the order is to group some of the multiplications together.\n\nLet me think about the possible groupings. Since we have four numbers: 2, 3, 4, 5, and three operations: multiplication, multiplication, and addition. To change the order, we can group some of the multiplications together.\n\nOne way is to group the first two numbers: (2·3)·4·5 + 1. Let me compute that: 2x3 is 6, then 6x4 is 24, 24x5 is 120, and 120 + 1 is 121. Wait, that's the same as before. So, that doesn't help.\n\nAnother way is to group the first three numbers: 2·(3·4·5) + 1. Let me see: 4x5 is 20, then 3x20 is 60, 60x2 is 120, and 120 + 1 is 121. Still 121. Hmm, so grouping the last two numbers doesn't change the result either.\n\nWait, maybe I should group the first two numbers and the second number together, but that didn't change anything. Wait, maybe grouping the first three numbers: 2·(3·4)·5 + 1. Wait, that's the same as before. Hmm.\n\nWait, perhaps I can group the first two numbers and the last two numbers separately? But since the addition is only between 5 and 1, I can't group 5 and 1 together without parentheses. So, maybe I can group 2·3·4·5 separately, but that's what I did before.\n\nWait, maybe I can group 2·3·4 and then multiply by 5, but that's the same as before.\n\nHmm, so maybe I can group 2·3·4·5 as a single term, but that would be 2x3x4x5, which is 120, and then add 1, which is 121. So, that doesn't change anything.\n\nWait, maybe I can group 2·(3·4·5) + 1. Let's compute: 3x4 is 12, and 12x5 is 60, 2x60 is 120, 120 + 1 is 121. Still 121.\n\nWait, so grouping the last two numbers are giving me 121. So, is 121 the only possible value? But the problem says \"how many values can be obtained,\" implying that there are multiple.\n\nWait, maybe I'm missing something. Let me think again.\n\nWait, maybe I can group the addition in a different way. But the addition is only between 5 and 1, so I can't group 5 and 1 together without parentheses. So, maybe I can't change the order of operations by grouping the addition.\n\nWait, but the problem says \"by inserting parentheses,\" so maybe I can group the entire expression in a different way. Let me try.\n\nWait, maybe I can group the entire expression as 2·3·4·5 + 1. So, if I group 2·3·4·5 as a single term, that's 2x3x4x5, which is 120, and then add 1, which is 121. So, that's the same as before.\n\nWait, maybe I can group 2·3·4·5 as (2·3)·(4·5) + 1, but that's the same result.\n\nWait, maybe I can group 2·3·4·5 as (2·3)·(4·5) + 1. Let's compute: 3x4 is 12, 12x5 is 60, 2x60 is 120, 120 + 1 is 121. Same result.\n\nWait, maybe I can group 2·3·4·5 as 2·(3·4)·5 + 1. Let's compute: 2x3 is 6, 6x4 is 24, 24x5 is 120, 120 + 1 = 121.\n\nSame result again.\n\nWait, maybe I can group 2·3·4·5 as 2·(3·4·5) + 1, which is the same as before.\n\nWait, maybe I can group 2·3·4·5 as 2·(3·4·5) + 1, which is the same as before.\n\nWait, so all these groupings are giving me 121. So, is 121 the only possible value? But the problem says \"how many values can be obtained,\" implying that there are multiple. Wait, maybe I'm missing something.\n\nLet me think again.\n\nWait, maybe I can group the addition in a different way. But the addition is only between 5 and 1, so I can't group 5 and 1 together without parentheses. So, maybe I can't change the order of operations by grouping the addition in a different way. But the problem says \"by inserting parentheses,\" so maybe I can group the entire expression in a different way. Let me try.\n\nWait, maybe I can group the entire expression as 2·3·4·5 + 1. So, if I group 2·3·4·5 as a single term, that's 2x3x4x5, which is 120, and then add 1, which is 121.\n\nSo, that's the same as before.\n\nWait, maybe I can group 2·3·4·5 as (2·3)·(4·5) + 1, which is 6x20 = 120 + 1 = 121.\n\n\n\nWait, same result.\n\nWait, maybe I can group 2·3·4·5 as 2·(3·4·5) +|CUT|✖

`Total: 2000+10 tokens`

---

**Comparison:**
Above are responses from two models—BudgetThinker and the Original model—to the same question under different budget constraints (500 and 2000).

It can be observed that under both budget conditions, BudgetThinker successfully completes its responses within the budget limits (409 and 1723 tokens, respectively) while perfectly adhering to the specified format. In contrast, the Original model exceeds the maximum token limit (500+10 tokens and 2000+10 tokens) and fails to extract the answer in the correct format.

BudgetThinker: Our BudgetThinker uses minimal steps to solve the problem. It identifies key properties for a multiple of 30 (divisibility by 2, 3, and 5) and constructs the smallest number directly. The number must end in 0 and have a digit sum that is a multiple of 3. Using three 2's (the smallest number of 2's to meet this condition), it constructs 2220. This direct approach is **concise and efficient**. The reasoning process is **succinct and clear,** starting **directly** from the properties of multiples of 30 and analyzing the unit digit and digit sum conditions step by step to quickly reach the conclusion. It fully considers that the multiple of 30 must meet the conditions of being a multiple of 2, 3, and 5, and closely revolves around these conditions in each step, **without any redundant thinking**, directly arriving at the smallest number that fits the criteria, 2220.

Original model: The original model is **less efficient**. It repeatedly verifies the same conditions and explores unnecessary possibilities after identifying 2220. This redundancy wastes tokens and leads to an incomplete, erroneous output due to exceeding the token limit. The thinking process is **lengthy and complex**, with repeated checks and validations, and even unnecessary repetitions and superfluous thoughts (such as considering whether there could be a smaller number, despite having already identified the smallest one). Although it also considers the conditions for a multiple of 30, the thinking process **lacks efficiency and does not directly use the known conditions to reach a conclusion**. Instead, it tries to find the answer through continuous attempts and validations, resulting in a lengthy and error-prone process.

Figure 8: This is another comparison between BudgetThinker and Original model's response cases on MATH datasets.

**Question:**
<image><image>
What happened between these two frames?
Choices:
A. Robot arm lifted the cup.
B. Robot arm poured all the nuts into a cup.
C. Robot arm poured some of the nuts into a cup.
D. Nothing happened.
Please answer directly with only the letter of the correct option and nothing else.Carefully analyze the multiple-choice question above and reason through it step by step.
Conclude your response with a line in the following format: $LETTER (without quotes), where $LETTER is the letter of the correct choice. \n(Complete thinking within 200 tokens or fewer)<think>

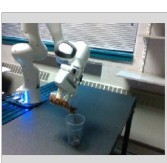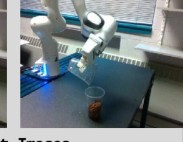
**Input Images**

**BudgetThinker:**
My visual sensors detect a robotic arm holding a transparent container filled with nuts. In the initial frame, the robotic arm is positioned above a transparent cup, suggesting the intent to transfer nuts into it. In the subsequent frame, the robotic arm appears to have tilted the container, indicating that nuts are being poured into the cup. The nuts\n<remaining>2/3</remaining>\n are visibly transferred from the container into the cup, with some nuts already visible in the cup from the first frame. This sequence of actions suggests that nuts are being poured into the cup, but not all nuts have been transferred yet. Comparing the options, (A) is incorrect as the robotic arm is still holding\n<remaining>1/3</remaining>\n the container, (B) is incorrect as not all nuts have been poured, (C) aligns with the visual evidence of nuts being poured into the cup, and (D) contradicts the observed actions. Therefore, the correct answer is (C).</think><answer>C</answer>✅
**Total: 191 tokens**

**Laser:**
My visual sensors detect a robotic arm holding a transparent container filled with nuts. In the initial frame, the robotic arm is positioned above a transparent cup, and the nuts are visible inside the container. In the subsequent frame, the robotic arm is still holding the container, but the nuts appear to have been partially transferred into the cup. The robotic arm's position remains consistent, indicating that the transfer process is ongoing.
The task is to determine what happened between these two frames. The visual evidence shows that nuts are being poured into the cup, but not all of them have been transferred yet. This suggests that the nuts were partially poured, not all of them. \n\nEvaluating the options: \n(A) \"Robot arm lifted the cup\" is incorrect as the cup remains stationary. \n(B) \"Robot arm poured all the nuts into a cup\" is incorrect as not all nuts have been poured. \n(C) \"Robot arm poured some of the nuts into a cup\" aligns with the visual evidence. \n(D) \"Nothing happened\"|CUT|✗
**Total: 200+10 tokens**

**Comparison:**
In this task, the thinking process had a 200-token budget. BudgetThinker used 191 tokens, delivering a complete and well-formatted answer. Laser exceeded the 210-token limit, causing truncated and erroneous output.
Analyze the two responses based on the following three dimensions:

**Efficient Reasoning:** BudgetThinker used a logical "observe→compare→judge" structure to identify key changes and eliminate wrong options efficiently. Laser lacked this framework, wasted tokens on incorrect options, and left its core argument unfinished.

**Budget Awareness:** By tracking usage with <remaining> tags, BudgetThinker dynamically adjusted its response. Laser showed no such awareness, over-describing minor details and rushing the critical reasoning.

**Precise Information:** BudgetThinker used only relevant visual evidence (e.g., "nuts not fully transferred") for concise arguments. Laser used full sentences for static details, creating a loose structure that failed to prioritize the core judgment.

**Question:**
<image>
What is about to happen here?
Choices:
A. Someone is reading the ingredients of the packet.
B. Someone is about to make tea, so they are tearing open a tea packet.
C. No action is depicted in this image.
D. Someone is trying to open a tea packet, and they are likely to destroy the teabag in the process of opening the package.
Please answer directly with only the letter of the correct option and nothing else.Carefully analyze the multiple-choice question above and reason through it step by step. Conclude your response with a line in the following format: $LETTER (without quotes), where $LETTER is the letter of the correct choice.
\n(Complete thinking within 80 tokens or fewer)<think>

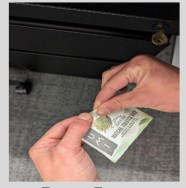
**Input Images**

**BudgetThinker:**
Visual sensors detect a hand gripping a small packet labeled \"Matcha Toasted Rice,\" with the thumb pressing down on\n<remaining>2/3</remaining>\n the corner. This suggests the packet is being opened to access its contents, likely a tea bag. The action aligns with the visual evidence of nuts being poured into to\n<remaining>1/3</remaining>\n make tea, as the packet is being opened to potentially damage the teabag.
</think><answer>D</answer>✅
**Total: 78 tokens**

**Laser:**
The image shows a hand holding a packet labeled \"Matcha Toasted Rice,\" with fingers positioned to tear it open. The packet is not being read or opened for tea-making. The action of tearing the packet suggests the intent to access its contents, likely for consumption.
Thus, the correct choice is (D) \"Someone is trying to open a tea packet, and they are likely to destroy the teabag in the process of opening the package|CUT|✗
**Total: 80+10 tokens**

**Comparison:**
In this task, the thinking process had a 80-token budget. BudgetThinker used 78 tokens, delivering a complete and well-formatted answer. Laser exceeded the 90-token limit, causing truncated and erroneous output.
Analyze the two responses based on the following three dimensions:

**Efficient Reasoning:** BudgetThinker uses a clear "Observation→Reasoning→Judgment" flow, linking visual cues (hand pinching tea bag) directly to Option D's logic (opening may damage the bag). Laser's structure is inconsistent, with contradictory claims (e.g., denies then affirms opening intent).

**Budget Awareness:** BudgetThinker tracks usage (<remaining>2/3</remaining>, <remaining>1/3</remaining>), ensuring full answer output (<answer>D</answer>) within limits. Laser wastes space on minor details, failing to output the complete <answer> tag before budget exhaustion.

**Precise Information:** BudgetThinker focuses on key evidence (e.g., thumb pressing corner) to support the "damage" conclusion. Laser adds irrelevant details (e.g., label content), weakening logical focus and precision.

Figure 9: This is a comparison between BudgetThinker and Laser's response cases on VQA datasets.

