# OpenReview forum: "BudgetThinker: Empowering Budget-aware LLM Reasoning with Control Tokens"
_ICLR.cc/2026/Conference — Submitted to ICLR 2026_

### Official Review · Reviewer_x15y · 2025-10-28

**Soundness:** 3
**Presentation:** 3
**Contribution:** 2
**Rating:** 4
**Confidence:** 3

**Summary:**

This paper presents a novel framework designed to empower LLMs with budget-aware reasoning. The method can provide precise control over the length of their thought processes. The main idea is to inserts special control tokens to inform the model of its remaining token budget. They conduct two training phases, first SFT then RL.

**Strengths:**

1. The paper focuses on an interesting question.
2. The paper is well-written and easy to follow.
3. The experiments show reasonable improvements.

**Weaknesses:**

1. The length setting appears to be manually selected. As shown in Table 1, the experiments were even evaluated under different length budgets for different dataset. However, users would not know the optimal length for each dataset in practice. Therefore, this setup seems somewhat unrealistic.
2. The paper mainly focuses on controlling the maximum output length. However, for relatively simple questions, forcing the model to produce unnecessarily long responses is neither efficient nor meaningful. It would be more realistic to allow adaptive length generation based on task difficulty.
3. As described in the paper, the proposed method appears to require huge computational resources for training. The SFT data generation process itself seems costly, and the full training pipeline involves both an SFT stage and two phases of RL. This raises concerns about the overall training efficiency and practicality of the approach.
4. I think some baselines are missing. Such as [1,2,3] all focus on reasoning with a length budget.
[1] Aggarwal, Pranjal, and Sean Welleck. "L1: Controlling how long a reasoning model thinks with reinforcement learning." arXiv preprint arXiv:2503.04697 (2025).
[2] Qi, Penghui, et al. "Optimizing anytime reasoning via budget relative policy optimization." arXiv preprint arXiv:2505.13438 (2025).
[3] Zhang, Xuechen, et al. "Making small language models efficient reasoners: Intervention, supervision, reinforcement." arXiv preprint arXiv:2505.07961 (2025).
[4] Li, Zhong-Zhi, et al. "TL; DR: Too Long, Do Re-weighting for Effcient LLM Reasoning Compression." arXiv preprint arXiv:2506.02678 (2025).

**Questions:**

1. How to choose the number of special control tokens K.
2. What would happen if the method were applied to a model that does not naturally produce overly long reasoning traces, and instead aims to explore longer reasoning budgets? Would the length constraint still provide benefits, or would it hinder the model’s ability to develop deeper reasoning skills?
3. For SFT, is it important to control the length distribution? Will it also help if the SFT data length distribution is very different from the length budget?
4. In the curriculum stage, the paper adopts a long-to-short training schedule. Moreover, how would the method adapt if the model’s current average generation length is already shorter than the budget?

---

> ### Author Response · Authors · 2025-11-21
>
> We thank Reviewer 4 for the constructive feedback. We reiterate that **BudgetThinker** is designed to provide precise length adherence for real-world "hard" constraints (e.g., strict latency limits). Below, we address the specific concerns with clarifications and new experimental evidence.
>
> # **Weakness 1: Practicality of Fixed Length Settings & Generalization**
> **Response:**
> We appreciate the feedback regarding the practicality of budget settings. We have clarified our design choice and provided new evidence of fine-grained generalization.
>
> * **Rationale for Fixed Intervals:** The budgets in Table 1 (e.g., 500, 1000... 10k) were selected specifically to align with reporting standards of baselines (e.g., Laser, ThinkPrune) to ensure fair comparison.
> * **Generalization to Unseen Budgets:** To prove BudgetThinker is not "overfitted" to specific training integers, we conducted new experiments on MATH-500 using a fine-grained budget spectrum (0.5k to 10k at 0.5k intervals). BudgetThinker demonstrates a smooth performance curve on "unseen" intermediate budgets.
>
> **Table R6: Fine-grained Budget Evaluation of 1.5B LLMs on MATH-500** ($B$ denotes reasoning token budget)
>
> | math500 | $B$=0.5k | 1k | 1.5k | 2k | 2.5k | 3k | 3.5k | 4k | 4.5k | 5k | 5.5k | 6k | 8k | 10k |
> | :--- | :--- | :--- | :--- | :--- | :--- | :--- | :--- | :--- | :--- | :--- | :--- | :--- | :--- | :--- |
> | Budgetthinker 1.5b | 59.4 | 75 | 77.6 | 81.6 | 82.4 | 84 | 84.6 | 84.8 | 84.4 | 85.8 | 83.8 | 85.8 | 87.2 | 87.4 |
> | anyreasoner | 51.0 | 66.8 | 74.2 | 76 | 79.2 | 80.6 | 81.8 | 83.4 | 84.2 | 84.4 | 84.2 | 84.8 | 86.4 | 87.4 |
> | Laser | 48.4 | 70.2 | 77.2 | 80.2 | 80.2 | 81.8 | 82.4 | 82.6 | 82.8 | 82.6 | 83.4 | 83.8 | 83.6 | 83.2 |
> | ThinkPrune | 44.0 | 69.4 | 77.6 | 79.4 | 79.8 | 80.4 | 83.4 | 82.8 | 81.2 | 80.6 | 80.2 | 83.3 | 81.2 | 83.6 |
>
> * **Real-World Application:** Regarding the concern that users "would not know the optimal length": In our target domain of controllable AI (e.g., real-time agents), the budget is also often determined by system constraints, such as **latency limits of the application, memory (RAM) constraints**, etc. rather than user estimation. In these cases, the system calculates the token limit corresponding to the remaining time (e.g., 100ms $\approx$ fixed token count), and our model adapts to that strict limit reliably.
>
> # **Weakness 2: Efficiency on Simple Questions vs. Adaptive Reasoning**
> **Response:**
> We agree that adaptive length is ideal for general chatbots, but we clarify that "Budget" in our framework acts as a **safety ceiling**, not a forced target.
>
> * **Hard Constraints:** In safety-critical scenarios (e.g., robotics), a model *must* conclude within a strict window. "Adaptive" methods that exceed this window cause system failure. BudgetThinker guarantees adherence.
> * **Budget as a Ceiling:** Our method does not force long responses for simple queries.
>     * **Evidence:** As shown in **Figure 2(b)**, when given a large budget ($B=10,000$) for simple tasks, the **Utilization Ratio drops to ~30-40%**.
>     * This proves the model naturally terminates generation once the solution is found, utilizing only a fraction of the available budget rather than "padding" the output.

---

> ### Author Response · Authors · 2025-11-21
>
> # **Weakness 3: Training Efficiency and Computational Cost**
> **Response:**
> Our analysis shows the total training cost is comparable to, or even lower than, standard baselines.
>
> * **High Sample Efficiency in RL:** Although we use a two-stage RL pipeline, the **Curriculum Strategy** accelerates convergence.
>     * **Standard RL:** Typically requires ~10 epochs to converge.
>     * **BudgetThinker:** Requires only 1 epoch per budget in the curriculum phase (4 total) + 1 epoch in the randomized phase. **Total: 5 epochs.**
>     * **Conclusion:** Our method achieves convergence with roughly **50% fewer epochs** than standard RL baselines.
> * **SFT Overhead:** The SFT process is computationally identical to standard "cold start" SFT. The offline data compression is a one-time curation step, similar to standard distillation pipelines, and the dataset is reusable across model sizes.
>
> ### **Weakness 4: Missing Baselines ([1]-[4])**
> **Response:**
> We thank the reviewer for identifying these recent works. We have expanded our comparison in the revision:
>
> * **Vs. Anytime Reasoning [2]:** We added a comparison in the above Table 6. While [2] focuses on arbitrary interruption, BudgetThinker achieves higher accuracy at specific targets because "Control Tokens" allow the model to *plan* its termination rather than being abruptly cut off.
>
> * **Vs. L1 [1]:** We prioritized comparing against **Laser**, a similar RL-based length control method. Since BudgetThinker significantly outperforms Laser (+4.9% avg. accuracy), and Laser is competitive with L1 (shown in the paper https://arxiv.org/pdf/2505.15612), this establishes our relative performance.
>
> * **Vs. Compression Methods [3, 4]:** We note a critical distinction: [3] and [4] focus on permanent *compression* (reducing redundancy). BudgetThinker is a **Budget-Aware** framework that can stretch *or* compress reasoning dynamically based on a user-defined variable $B$.

---

> ### Author Response · Authors · 2025-11-21
>
> # **Question 1: Strategy for Choosing Control Token Number ($K$)**
> **Response:**
> The choice of $K$ balances **Guidance Granularity** vs. **Reasoning Continuity**.
>
> * **Signal-to-Noise Trade-off:**
>     * **Too Frequent (e.g., every 50 tokens):** Disrupts semantic flow and degrades reasoning (see Figure 3a).
>     * **Too Sparse:** The model lacks sufficient feedback to correct its pacing, leading to interrupted reasoning.
> * **Optimal Setting:** We found **$K=8$** is the "sweet spot" for text reasoning (budgets 500-10k). It creates manageable planning blocks. For MLLMs with naturally shorter horizons, we adjust to $K=3$.
>
> # **Question 2: Applicability to Models Exploring Longer Reasoning**
> **Response:**
> We thank the reviewer for this insightful question regarding the generalizability of our framework to models seeking to extend, rather than compress, their reasoning.
> We believe that BudgetThinker would not hinder the development of deeper reasoning skills; rather, it provides a structured mechanism for reasoning planning. We offer two perspectives to support this:
>
> * **Budget as a "Pacing" Mechanism:** While our experiments primarily highlight efficiency (managing strict constraints), the core contribution of BudgetThinker is controllability, not merely truncation. For a model attempting to explore longer reasoning paths, we can first set a high budget to let it explore reasoning space.
> * **Reinforcement Learning Prioritizes Accuracy:** As noted in recent literature (e.g., Kimi k1.5 https://arxiv.org/abs/2501.12599), even models designed for reasoning benefit from length-relative penalties to prevent infinite loops or meaningless repetition. Our reward function (Eq. 2 in Section 2.4) balances correctness ($k_1$) and length adherence ($k_3$). Crucially, the correctness reward is dominant. If "deeper reasoning" is required to solve a complex problem, the RL process drives the model to allocate its budget toward those high-value reasoning steps.
>
> # **Question 3: Importance of SFT Length Distribution**
> **Response:**
> Empirical results show that **maintaining a balanced length distribution during SFT is critical**.
>
> * **Avoiding Overfitting:** If the SFT data is heavily biased toward a specific length (e.g., short reasoning), the model tends to overfit to that reasoning depth, limiting its ability to generalize to other budgets.
> * **Ablation Study:**
>     - **Original settings in the paper (Balanced):** As described in the paper, we constructed a mixture of long reasoning data (e.g., Bespoke-Stratos-17k, s1k), compressed long reasoning data, and short reasoning data (e.g., NuminaMath). Crucially, since NuminaMath contains a massive number of short samples (~859k), we downsampled it (to ~20k) and augmented the long data to ensure a uniform distribution across reasoning lengths.
>     - **Ablation Study (Balanced vs. Biased):** To quantify this benefit, we conducted an ablation study where we add all the NuminaMath data to the dataset, leading to a biased distribution that has too many short reasoning data. We trained a "Biased" model using the full, raw dataset, which is heavily dominated by short reasoning samples from NuminaMath. The results (Table R7) demonstrate a clear performance degradation in the "Biased" model, particularly at higher budgets:
>
> ### Table R7: Balanced vs. Biased Dataset Performance of BudgetThinker-1.5B on MATH-500 ($B$ denotes reasoning token budget)
>
> | Math500-1.5B | $B$=500 | $B$=1000 | $B$=2000 | $B$=4000 | $B$=6000 | $B$=10000 |
> | :--- | :--- | :--- | :--- | :--- | :--- | :--- |
> | Balanced | 62.4 | 69.6 | 81.2 | 83 | 85 | 84.2 |
> | Biased | 62.6 | 67.2 | 75.2 | 80.6 | 79.2 | 80.6 |
>
>
>
> # **Question 4: Adaptation with Long-to-Short Curriculum**
> **Response:**
> * **Curriculum Rationale:** Starting with large budgets allows the model to learn the semantic meaning of Control Tokens under loose constraints before facing tight restrictions.
> * **Adaptation (Natural Length < Budget):** As discussed in Weakness 2, the budget $B$ is a safety ceiling. If the natural reasoning path is shorter than $B$, the control tokens signal ample remaining capacity. The model, driven by the correctness reward, will output the final answer immediately rather than wasting tokens, ensuring efficiency naturally.

---

### Official Review · Reviewer_Vx6G · 2025-10-29

**Soundness:** 2
**Presentation:** 3
**Contribution:** 2
**Rating:** 2
**Confidence:** 4

**Summary:**

This paper introduces BudgetThinker, a framework designed to give Large Language Models (LLMs) and Multimodal LLMs (MLLMs) precise control over the length of their Chain-of-Thought (CoT) reasoning processes. The core innovation is the periodic insertion of special control tokens during inference, which continuously inform the model of its remaining token budget. This is supported by a comprehensive two-stage training pipeline: 1) Supervised Fine-Tuning (SFT) on a curated dataset that includes these control tokens and spans a wide range of reasoning lengths, and 2) Reinforcement Learning (RL) using a curriculum and a length-aware reward function to optimize for both accuracy and strict budget adherence. Experiments on mathematical and multimodal reasoning benchmarks demonstrate that BudgetThinker achieves a better trade-off, maintaining high accuracy while precisely following specified token budgets, outperforming several baselines.

**Strengths:**

* The use of the periodic special tokens to remind the LLM its remaining budget is somewhat novel, and incorporating it into SFT and RL is reasonable.
* Experiments cover both LLM and VLM.

**Weaknesses:**

* The paper shows that the model adheres to the budget, but offers less insight into how the reasoning strategy changes. A qualitative analysis comparing the CoT traces at different budgets (e.g., does it skip certain reasoning steps? does it use more abbreviations?) would be very valuable.
* The method is a bit complex and costly due to the SFT stage.

**Questions:**

* When conducting GRPO, we assume the data is sampled from \pi_old and we compute the token ratio using \pi_\theta and \pi_\old. In this method, since you insert special tokens while sampling, you essentially change the distribution of \pi_old. How did you deal with the GRPO loss computation?

* I recently came across this paper Optimizing Anytime Reasoning via Budget Relative Policy Optimization (https://openreview.net/pdf/8136e4668a09f8c47a2454d9e72728d4fdea055e.pdf), which shares very similar motivation (their paragraph 2 v.s. your paragraph 2) and method (they truncate at some sampled budget while you insert at the budgeted locations). It would be nice if you can discuss your differences and how your method is fundamentally better than the baseline.

---

> ### Author Response · Authors · 2025-11-21
>
> ## Weakness 1: Qualitative Analysis
> **Response:**
>
> We thank the reviewer for suggesting this valuable qualitative analysis. We have provided some examples in **Appendix A.7**, and now we expand it with a new case study (illustrated in **Figure 9** of the revision) that traces the reasoning evolution for a single hard problem across budgets ranging from 500 to 6000 tokens.
>
> **Qualitative Analysis of Reasoning Strategy**
> As summarized in our new analysis, the model does not merely truncate text; it fundamentally restructures its problem-solving approach:
>
> 1.  **Reasoning Depth (Step Count):**
>     * **Tight Budget ($B=500$):** The model adopts a "Direct Solver" persona. It condenses the process into just **3 primary steps**, skipping theoretical framing to jump straight to the most efficient solution method (e.g., "Reformulate... Enumerate").
>     * **Large Budget ($B=6000$):** The model acts as a "Comprehensive Explorer," expanding to **13 steps**. It includes pedagogical elements like "Explain the meaning of indistinguishable" and explores theoretical connections (e.g., "generating function verification") that are absent at lower budgets.
>
> 2.  **Verification Intensity:**
>     * **Tight Budget:** Verification is "Minimal"—often a single, rapid check (e.g., "one box containing 5...") to ensure basic consistency.
>     * **Large Budget:** Verification becomes "Very extensive." The model employs **multiple distinct methods** to cross-validate the answer (e.g., algebraic equations vs. manual enumeration vs. standard formulae) and runs specific sanity checks (e.g., $n=1,2,3,4$).
>
> 3.  **Syntactic Compression:**
>     At lower budgets, the model uses concise, declarative sentences and immediate calculations. At higher budgets, it engages in "meta-reasoning," explicitly discussing why certain approaches work (e.g., "Explain why indistinguishability collapses ordered solutions") and reiterating conclusions for emphasis.
>
> **Conclusion**
> This analysis confirms that BudgetThinker learns a **strategic trade-off**: it sacrifices theoretical depth and redundancy for speed when necessary, but autonomously recovers these rigorous behaviors when the budget permits.
>
> ## Weakness 2: Complexity/Cost of SFT
> **Response:**
> We clarify that the SFT stage is minimally costly and actually **accelerates** the overall pipeline.
> * **Low Cost:** The SFT process is equivalent to a standard "cold start" fine-tuning (taking only ~5 hours on 8xA800 for 7B). The token insertion adds negligible overhead.
> * **SFT Accelerates RL:** By teaching the semantics of Control Tokens during SFT, we "cold start" the RL process. This makes the expensive RL phase converge much faster (requiring only ~5 epochs total) compared to training from scratch.

---

> ### Author Response · Authors · 2025-11-21
>
> # Question 1: GRPO Mathematical Details
>
> **Response:**
> Our method remains mathematically consistent with standard GRPO. We treat special control tokens as part of the **environment state (context)** rather than model **actions**.
> * **Masking:** The special tokens are fixed inputs; their loss is masked out (similar to user prompt tokens).
> * **Consistency:** Since $\pi_{old}$ and $\pi_{\theta}$ both see the *same* special tokens at the *same* positions (insertion is deterministic based on length), the probability distributions remain comparable.
> * **Stability:** Because $\pi_{old}$ is fine-tuned via SFT to understand these tokens, they function as informative state signals rather than out-of-distribution noise, ensuring stable convergence.
>
> # Question 2: Comparison with "Anytime Reasoning" (BRPO)
>
> **Response:**
> We have added a comparison with BRPO (AnyReasoner).
> * **Fundamental Difference:** BRPO addresses **Anytime Interruption** (optimizing for random truncation), while BudgetThinker addresses **Budget-Aware Planning**.
> * **Passive vs. Active:** BRPO models optimize for "survival" (maximizing accuracy at every step). BudgetThinker uses Control Tokens for **Active Planning**, allowing it to confidently embark on complex reasoning paths when capacity permits.
> * **Result:** As shown below, BudgetThinker generalizes better to specific fine-grained budgets.
>
> **Table R5: Fine-Grained Budget Evaluation on Math500-1.5B** ($B$ denotes reasoning token budget)
>
> | Method | $B$=500 | $B$=1000 | $B$=2000 | $B$=2500 | $B$=3500 | $B$=4000 | $B$=6000 | $B$=10000 |
> | :--- | :--- | :--- | :--- | :--- | :--- | :--- | :--- | :--- |
> | BudgetThinker | 59.4 | 75.0 | 81.6 | 82.4 | 84.6 | 84.8 | 85.8 | 87.4 |
> | AnyReasoner | 51.0 | 66.8 | 76.0 | 79.2 | 81.8 | 83.4 | 84.8 | 87.4 |
> | Laser | 48.4 | 70.2 | 80.2 | 80.2 | 82.4 | 82.6 | 83.4 | 83.2 |

---

### Official Review · Reviewer_rR7t · 2025-11-02

**Soundness:** 3
**Presentation:** 3
**Contribution:** 3
**Rating:** 6
**Confidence:** 2

**Summary:**

This paper presents BudgetThinker, a framework that enables LLMs to perform budget-aware reasoning by periodically inserting special control tokens during inference. The approach consists of two key components: (1) a ratio-based control token insertion strategy using a fixed set of K=8 tokens to signal remaining budget fractions, and (2) a comprehensive two-stage training pipeline combining supervised fine-tuning with curriculum-based reinforcement learning that incorporates a length-aware reward function. The authors demonstrate improvements over strong baselines (ThinkPrune, Laser) on mathematical benchmarks (MATH-500, AMC 2023) and multimodal reasoning tasks (EQRA, SAT), showing an average accuracy improvement of 4.9% across various budgets while maintaining better budget adherence.

**Strengths:**

Originality:
- The ratio-based control token design is elegant and scalable, requiring only K fixed tokens regardless of budget size, which addresses a practical limitation of fixed-interval approaches
- The training methodology is thoughtfully designed, with a well-motivated balance of three data types (long reasoning, iteratively compressed, non-thinking) and a sensible curriculum learning strategy
- The plug-and-play nature of the framework makes it potentially valuable as a complementary module for existing test-time scaling methods

Quality:
- The experimental evaluation spans multiple model scales (1.5B, 7B), modalities (text-only and vision-language), and task types, demonstrating breadth
- The ablation studies systematically examine key design choices (control token strategies, iterative training, RL contribution), providing insight into what drives performance
- Figure 2 effectively demonstrates the practical advantages in both budget following and budget utilization compared to baselines

Clarity:
- The motivation is well-articulated, clearly establishing the need for precise budget control in latency-critical applications
- The method description flows logically from token insertion mechanics through data curation to RL training
- The visualizations, particularly Figure 1, help readers quickly grasp the core idea

Significance:
- The work addresses a genuine practical need for deploying reasoning models in resource-constrained settings
- The evaluation on both mathematical and multimodal reasoning demonstrates applicability beyond a single domain
- The improved budget adherence could be particularly valuable for production systems with hard latency requirements

**Weaknesses:**

- The task coverage is somewhat focused on mathematical reasoning. I'm interested in how the approach might generalize to other important applications like code generation, open-domain QA, or multi-step agentic tasks
- For the 7B experiments, it would be helpful to include ThinkPrune comparisons if feasible
- The budget range tested (50-10K tokens) is reasonable, though I wonder about behavior in more extreme regimes
Beyond Pass@1 accuracy, it might be illuminating to examine additional dimensions like reasoning quality, consistency, or calibration

**Questions:**

1. Task generalization: I'd be interested to know how the approach performs on:
- Code generation (where completeness vs. brevity tradeoffs differ)
- Open-ended QA (without clear correct answers)
- Multi-step agent tasks (requiring planning + execution)

2. Failure mode characterization:
- Which problem types are most challenging for budget control? Could you provide quantitative analysis?
- Figure 2 shows lower budget following ratios at small budgets for BudgetThinker—what drives this behavior?

3. Alignment with human preferences:
- Have you conducted human evaluations? Do users always prefer shorter reasoning?
- In some contexts, detailed reasoning might be valuable for trust and debugging—how does the framework accommodate this?

---

> ### Author Response · Authors · 2025-11-21
>
> ## Weakness 1 & Question 1: Task Generalization (Code, QA, Agents)
> **Response:**
> We agree that generalization is crucial. While our current work features math, we wish to highlight that our evaluation **already extends to Multimodal and Embodied Reasoning**, which serves as a strong proxy for agentic capabilities.
>
> * **Multimodal & Spatial Reasoning:** Table 1 and Figure 5 evaluate BudgetThinker on **ERQA** (Embodied Reasoning QA) and **SAT** (Spatial Aptitude Training). These tasks require the "Observe $\to$ Reason $\to$ Act" loop foundational to agentic tasks.
> * **Performance:** BudgetThinker achieves state-of-the-art results (surpassing Laser by ~3.4% on SAT), proving efficacy in grounded, physical-world reasoning.
> * **Dimensions of Generalization:** We have demonstrated generalization across **Modality** (Text/Vision), **Structure** (Symbolic/Spatial), and **Budget** (Long/Short). This strongly suggests the underlying mechanism—active planning via control tokens—would naturally extend to agentic and QA, which we will evaluate in the final paper.
>
> ## Weakness 2: 7B Model Comparisons
> **Response:**
> We noted in the manuscript that the official ThinkPrune repository lacks a 7B model. We **have trained a ThinkPrune-7B model from scratch** using their official methodology and our training data, and will include these results in the final version to ensure robust benchmarking, the results of Math-500 are shown below, where BudgetThinker-7B is still better than ThinkPrune-7B under most budgets.
>
> | Method | $B$=500 | $B$=1000 | $B$=2000 | $B$=4000 | $B$=6000 | $B$=10000 |
> | :--- | :--- | :--- | :--- | :--- | :--- | :--- |
> | **BudgetThinker-7B** | 69.4 | 77 | 86 | 90 | 92.2 | 93 |
> | **Thinkprune-7B** | 52.8 | 75.4 | 85.6 | 90 | 90.6 | 91.6 |

---

> ### Author Response · Authors · 2025-11-21
>
> ## Question 2: Failure Mode Characterization
> **Response：**
>
> We thank the reviewer for this insightful question regarding failure modes. To characterize the challenges of budget control, we conducted a quantitative analysis across different mathematical categories on MATH-500 (summarized in **Table R4**).
>
> **1. Quantitative Analysis: Challenging Problem Types**
>
> **Table R4: Performance Breakdown by Math Category (Math500-1.5B)**
>
> | Math Category | Average Accuracy (%) | Budget Following (%) |
> | :--- | :--- | :--- |
> | **Sequences / Recurrence** | 67.31 | 81.92 |
> | **Geometry** | 70.4 | 83.08 |
> | **Probability** | 75.11 | 86.43 |
> | **Logic / Puzzle** | 77.4 | 78.85 |
> | Algebra | 83.91 | 88.41 |
> | Counting & Combinatorics | 87.84 | 93.55 |
> | Word Problems (GSM8K-like) | 89.74 | 91.28 |
> | Functions & Graphing | 92.11 | 92.9 |
> | Arithmetic / Number Theory | 94.49 | 91.35 |
>
>
> **Insight:** We observe a clear correlation between **structural complexity** and budget adherence difficulties:
>
> * **High Challenge (Logic, Sequences, Geometry):** These categories exhibit the lowest adherence (e.g., Logic/Puzzle at **78.85%**). The primary driver is **Irreducible Complexity**:
>     * *Sequences/Recurrence:* These require strict iterative dependencies (step $n$ relies on step $n-1$). Skipping steps to save tokens often breaks the proof chain.
>     * *Logic/Puzzle:* These often require exhaustive case enumeration or complex conditional branching which cannot be compressed without missing critical logical paths.
> * **High Control (Arithmetic, Functions):** Categories like *Functions & Graphing* achieve high adherence (**92.9%**). These problems typically follow linear, formulaic paths where intermediate calculation steps can be easily merged or compressed without losing logical coherence.
>
> **2. Explanation for Lower Adherence at Small Budgets (Figure 2)**
>
> The lower adherence at strict budgets (e.g., 500 tokens) is driven by the tension between **Hard Constraints** and **Solvability**.
> * **Intrinsic Minimum:** Certain complex problems possess a physical minimum length required to derive the correct answer.
> * **Adaptive Trade-off:** When the budget is below this intrinsic minimum, BudgetThinker prioritizes **completeness over strict truncation**. The model learns to commit a "soft violation" (slightly exceeding the limit to finish the logic) rather than halting abruptly, which would guarantee an incorrect answer. This behavior effectively maximizes the expected utility of the response.
>
>
> ## Question 3: Alignment with human preferences
>
> **Response:**
>
> We appreciate the reviewer’s question regarding human alignment. While large-scale human evaluation of complex reasoning chains is challenging, we address these concerns through extensive **qualitative analysis (Appendix A.7)** and by highlighting the inherent **adaptability** of our framework.
>
> **1. Qualitative Evaluation of Budget-Aware Reasoning**
>
> In **Appendix A.7**, we performed a deep qualitative assessment to understand *how* the model adapts to strict budgets. We found that BudgetThinker does not simply truncate thoughts; instead, it adopts specific strategies to maintain **completeness**:
>
> * **Syntactic Compression:** Switching from verbose descriptions to concise, evidence-based statements.
> * **Skipping Redundancy:** Adopting a "direct approach" by identifying key properties immediately rather than exploring unnecessary possibilities.
> * **Reduced Verification:** Minimizing repetitive self-correction steps that are common in the original model.
>
> While these strategies may theoretically introduce a margin for error (calibration risk), they ensure the reasoning chain remains **logically complete** and syntactically valid, which we argue is preferable to the "truncated and erroneous output" often produced by baselines under pressure.
>
> **2. Accommodating Trust and Debugging**
> Regarding trust and debugging, our framework inherently accommodates these needs through the flexible budget parameter, allowing users to simply assign high budgets (e.g., $B=10,000$) to elicit detailed, exhaustive reasoning whenever verification is required.

---

### Official Review · Reviewer_NBim · 2025-11-08

**Soundness:** 2
**Presentation:** 2
**Contribution:** 2
**Rating:** 6
**Confidence:** 3

**Summary:**

The paper proposes budget thinking, a combination of test time control token insertion and RL training reward that teaches the model to make use of its test time budget. Across the board it seems that budget thinking is better at trading off compute vs. performance and controls token generation at run time.

**Strengths:**

The paper is interesting in introducing a combo of test time behavior and RL training reward to better elicit token control behavior for length-controlled generation. The method sounds novel and has rather solid tech contributions.

**Weaknesses:**

One weakness is that comparison against baseline seems relatively weak in that there is no alternative RL training method that gets compared, and it will be useful to have extra ablations to better understand the mechanism of the improvement.

**Questions:**

=== *budget think with no training* ====

I think it's useful to compare the current method vs. a baseline that just inserts the control token at test time - without training, would the model be able to already in-context take the hint and follow instructions better? This comparison will highlight whether there is a genuine gain in training model against such behavior vs. just in-context hint.

=== *baseline comparison in Fig 2* ====

I think the flagship comparison in Fig 2 shows improvements from budget thinker but it seems that the pruning think baseline does not make use of extra RL training? So this is a comparison between test time algorithm vs. test + train time algorithm. If the proposed method is quite novel and I understand there is no similar training in the literature yet, maybe we can ablate various ways that the reward function is set up to better understand the performance gains.

One idea is to train the system using typical reward but with insertion of the control token - this will essentially not penalize the model at training time when it violates the token budget. This comparison will show whether penalizing the token budget violation at RL time is necessary, or it suffices to just prompt the model + let it in-context learn. My hunch is that if we train long time without penalizing the violation, the model will learn to ignore such control tokens and the violation rate goes up over time, but it's good to verify if possible.

==== *SFT vs RL* ====

Fig 3 shows limited gain of using RL vs. SFT in combination with budget thinking, is that right? Does that mean that budget thinking constrains the RL performance vs. SFT, how about we just do regular RL vs. SFT and we should typically expect much larger gains from RL?

---

> ### Author Response · Authors · 2025-11-21
>
> ## Weakness 1: Weak Baselines & Need for Mechanism Analysis
>
> **Response:**
> We thank the reviewer for their valuable feedback on our experimental design. We clarify the selection of our baselines and highlight the ablation studies that isolate the improvement mechanism.
>
> **1. Clarification on RL Baselines**
> We respectfully clarify that our baselines, **ThinkPrune** and **Laser**, are fundamentally **RL-based reasoning optimization methods**:
>
> * **ThinkPrune (Hou et al., 2025):** As noted in Section 3.1, this method explicitly "uses GRPO for budget control" to prune reasoning traces.
>
> * **Laser (Liu et al., 2025):** This method employs "adaptive length-based reward shaping" within a reinforcement learning framework.
>
> Therefore, Table 1 in the main paper already provides a systematic comparison against the most representative alternative RL training pipelines. **BudgetThinker** outperforms these RL methods by an average of **4.9%**, demonstrating the superiority of our control-token-based method over standard RL pruning or reward shaping.
>
> **2. Mechanism Analysis (Ablations)**
> To address the request for "extra ablations," we highlight the detailed component analyses in **Sections 3.3 and 3.5**, where we analyzed the Role of **Control Tokens (vs. Pure RL)** and **RL Training (vs. Pure SFT)**.
> Here, we provide an **Additional Ablation**, where we include an **"RL-from-scratch" (Skip SFT)** ablation below. This demonstrates that SFT is critical for "grounding" the semantic meaning of control tokens before RL optimization begins.
>
> **Table R1: Impact of SFT Initialization on Math500-1.5B** ($B$ denotes reasoning token budget)
>
> | Method | $B$=500 | $B$=1000 | $B$=2000 | $B$=4000 | $B$=6000 | $B$=10000 |
> | :--- | :--- | :--- | :--- | :--- | :--- | :--- |
> | SFT+RL (Ours) | 59.4 | 75.0 | 81.6 | 84.8 | 85.8 | 87.4 |
> | Only RL | 55.7 | 68.3 | 77.1 | 80.5 | 82.8 | 83.0 |

---

> ### Author Response · Authors · 2025-11-21
>
> ## Question 1: Performance Without Training (Inference-Only)
>
> **Response:**
> We thank the reviewer for this valuable suggestion. Distinguishing between "training-induced" capabilities and "in-context" capabilities is essential.
>
> **1. Existing Training-Free Baselines**
> Our paper already includes strong training-free baselines in **Table 1**:
>
> * **Chain of Draft (CoD), c2f, and Original Model:** These rely entirely on in-context instructions (e.g., "compress reasoning", "answer briefly") without parameter updates.
> * **Results:** As shown in Table 1, these methods significantly underperform BudgetThinker. This empirically demonstrates that "in-context hints" alone are insufficient for stable budget adherence.
>
> **2. Requested Ablation: No-Training Control Tokens**
> To directly address the specific proposal, we tested a **"Countdown Token"** baseline where we insert our control tokens into the *original* model during inference without training.
>
> * **Results:** As shown below, this baseline performs worse than BudgetThinker. Without the SFT stage to ground the semantic meaning, the original model perceives these tokens as out-of-distribution noise.
>
> **Table R2: Training-Free vs. Trained Control Tokens on Math500-1.5B** ($B$ denotes reasoning token budget)
>
> | Method | $B$=500 | $B$=1000 | $B$=2000 | $B$=4000 | $B$=6000 | $B$=10000 |
> | :--- | :--- | :--- | :--- | :--- | :--- | :--- |
> | BudgetThinker | 59.4 | 75.0 | 81.6 | 84.8 | 85.8 | 87.4 |
> | Countdown Token (No Train) | 53.4 | 67.4 | 74.4 | 79.2 | 78.4 | 78.6 |
>
>
> > **Conclusion:** The performance gain stems from the **SFT+RL pipeline**, which **allows the model to learn the *semantics* of budget signals and the *strategy* of pacing—capabilities that do not emerge from test-time hints alone.**
>
> ---
>
> ## Question 2: Baseline Comparison & Reward Function Ablation
>
> **Response:**
> **1. Clarification: Baselines are RL Methods**
> We respectfully clarify that both **ThinkPrune** and **Laser** in Figure 2 are **training-based RL methods**, not test-time heuristics. ThinkPrune uses GRPO for pruning, and Laser uses RL with reward shaping. We will explicitly highlight this in the final manuscript.
>
> **2. Requested Ablation: Training with Control Tokens but NO Length Penalty**
> We conducted the suggested ablation: training with Control Tokens inserted, but using a standard reward function (Accuracy only) without the length-violation penalty.
> * **Model Ignores Tokens:** Without the penalty, the model learns to **ignore the control tokens**. The RL process treats them as noise to be filtered out.
> * **Performance Degradation:** The "SFT+RL (without length reward)" model performs significantly worse (82.2% vs 87.4% at high budgets) and even underperforms the SFT-only baseline. This confirms that the **length-aware penalty** is necessary to force the model to semantically ground the control tokens to reasoning length.
>
> | Method | $B$=500 | $B$=1000 | $B$=2000 | $B$=4000 | $B$=6000 | $B$=10000 |
> | :--- | :--- | :--- | :--- | :--- | :--- | :--- |
> | BudgetThinker | 59.4 | 75.0 | 81.6 | 84.8 | 85.8 | 87.4 |
> | SFT-only | 62.4 | 69.6 | 81.2 | 83.0 | 85.0 | 84.2 |
> | SFT+RL (without length reward) | 65.0 | 68.2 | 77.4 | 83.0 | 82.2 | 82.2 |
>
> ---
>
>
> ## Question 3: SFT vs. RL Gains
>
> **Response:**
> We respectfully clarify that RL gains are different under different budgets, and our method actually **enables** RL to succeed under strict budgets where "Regular RL" fails.
>
> **Why Gains are Small at Low Budgets: **
> Reviewer correctly noted "limited gains" at lower budgets (e.g., 1k). This is a budget-dependent phenomenon, not a flaw of our method: At tight budgets, the "solution space" is narrow. The model is constrained to simply "finishing the answer" (Imitation/SFT is sufficient), leaving little room for RL to explore diverse reasoning paths.
>
> To prove this, we implement an ablation, where we remove the control tokens (No-CT), and compare the performance of SFT-only and SFT+RL. As seen in the tables, neither BudgetThinker nor Regular RL shows large gains over SFT at 1k/2k budgets. However, as the budget opens up (4k+), the gap widens, and BudgetThinker's advantage becomes dominant.
>
> **Table R3-1: Ablation of Control Tokens on AMC23-1.5B (Accuracy %)** ($B$ denotes reasoning token budget)
>
> | Method | $B$=1000 | $B$=2000 | $B$=4000 | $B$=6000 | $B$=10000 |
> | :--- | :--- | :--- | :--- | :--- | :--- |
> | **BudgetThinker** | 46.2 | 65.5 | 70.4 | 72.5 | 74.8 |
> | No-CT SFT | 52.5 | 55.0 | 62.5 | 67.5 | 70.0 |
> | No-CT SFT+RL | 47.5 | 61.1 | 70.1 | 71.9 | 70.4 |
>
> **Table R3-2: Ablation of Control Tokens on Math500-1.5B (Accuracy %)** ($B$ denotes reasoning token budget)
>
> | Method | $B$=1000 | $B$=2000 | $B$=4000 | $B$=6000 | $B$=10000 |
> | :--- | :--- | :--- | :--- | :--- | :--- |
> | **BudgetThinker** | 75.0 | 81.6 | 84.8 | 85.8 | 87.4 |
> | No-CT SFT | 68.2 | 80.2 | 83.0 | 83.6 | 84.8 |
> | No-CT SFT+RL | 71.8 | 77.6 | 83.8 | 83.6 | 82.8 |

---

### Author Response · Authors · 2025-12-04
**Summary of Rebuttal**

First, we sincerely thank the newly assigned Area Chair for their time and effort. To help save your time, we have summarized our rebuttal and the new experiments we completed below.

**1. Strengths**

We are encouraged that the reviewers found common ground on the key contributions of BudgetThinker:
* **Originality & Novelty:** The novel combination of periodic control token insertion with a "thoughtfully designed" SFT and RL pipeline.
* **Scalability & Design:** The ratio-based token strategy was highlighted as "elegant and scalable," addressing the limitations of fixed-interval approaches.
* **Evaluation:** The experiments covered both LLMs and VLMs, demonstrating the framework's versatility.
* **Practicality:** The method addresses a "genuine practical need" for deploying reasoning models in resource-constrained and latency-critical environments.

**2. Summary of Extensive New Experiments**

During the rebuttal period, we performed comprehensive additional experiments to address every specific concern raised by the reviewers. We believe these results provide robust evidence to resolve pending questions:

* **Robustness against Baselines:**
    * **7B Model Comparison:** We trained **ThinkPrune-7B** from scratch to provide a direct comparison at the 7B scale. Results show BudgetThinker consistently outperforms ThinkPrune-7B across most budgets.
    * **Other Baselines:** We added comparisons against **AnyReasoner (BRPO)**, demonstrating that BudgetThinker achieves higher accuracy through active planning rather than passive interruption. We also addressed comparisons with **L1** and other compression methods.
* **Mechanism & Ablation Studies:**
    * **Training-Free Baseline:** We proved that simple "inference-only" token insertion (without training) fails, confirming the necessity of our SFT+RL pipeline.
    * **Reward Function Analysis:** We demonstrated that the length-aware penalty in RL is critical; removing it causes the model to ignore control tokens.
    * **SFT Data Distribution:** We conducted ablations showing that a balanced length distribution in SFT is essential for preventing overfitting to short reasoning.
* **Deep Analysis of Reasoning:**
    * **Qualitative Evolution:** We added a detailed case study (Figure 9 in the revision) tracing how the model shifts strategies from a "Direct Solver" (at strict budgets) to a "Comprehensive Explorer" (at high budgets), proving it does not just truncate text but restructures reasoning.
    * **Fine-Grained Generalization:** We tested the model on "unseen" intermediate budgets (e.g., 2.5k, 3.5k), showing a smooth performance curve and proving the model is not overfitted to specific training integers.

**3. Response to Reviewer Vx6G**

We have fully addressed the two specific concerns raised by Reviewer Vx6G:
* **Concern 1 (Qualitative Analysis):** We provided the requested qualitative analysis (detailed in Appendix A.7 of the revision).
* **Concern 2 (Training Cost):** We clarified that the SFT stage is computationally inexpensive (equivalent to a standard "cold start") and actually *accelerates* the subsequent RL convergence, making the pipeline highly efficient.
* **Concern 3 (GRPO Mathematical Details):** Our method remains mathematically consistent with standard GRPO. We treat special control tokens as part of the environment state (context) rather than model actions.

Reviewer Vx6G explicitly acknowledged the novelty of our method and the breadth of our evaluation. As the rejection recommendation was primarily based on the concerns listed above—all of which have now been resolved with new data and clarifications—we respectfully suggest that the grounds for the low score have been effectively removed.

**4. Conclusion**
We are confident that BudgetThinker offers high value to the community by introducing a new method to solve the problem of precise reasoning under strict constraints. We hope the Area Chair will re-evaluate the submission in light of these substantial improvements.

Thank you for your time and consideration.

**References:**

[1] Bairu Hou, Yang Zhang, Jiabao Ji, Yujian Liu, Kaizhi Qian, Jacob Andreas, and Shiyu Chang. Thinkprune: Pruning long chain-of-thought of llms via reinforcement learning, 2025. URL https://arxiv.org/abs/2504.01296.

[2] Penghui Qi, Zichen Liu, Tianyu Pang, Chao Du, Wee Sun Lee, and Min Lin. Optimizing Anytime Reasoning via Budget Relative Policy Optimization, 2nd AI for Math Workshop @ ICML 2025. URL https://openreview.net/forum?id=kMGc0yvM61.

[3] Pranjal Aggarwal and Sean Welleck. L1: Controlling how long a reasoning model thinks with reinforcement learning, 2025. URL https://arxiv.org/abs/2503.04697.

---

### Meta-Review · Area_Chair_cYVE · 2025-12-14

**Summary:**

The paper addresses budget aware LLM reasoning. The main innovation is to additionally (other than indicating the budget in the prompt) insert control tokens periodically to indicate the remaining budget. The novelty of this is somewhat limited, although the experiments show its effectiveness. The rebuttal provided reasonable clarification to most questions that the reviewers had.

**Reviewer Concerns:**

Reviewer Nbim commented that the baselines are weak with no RL methods and the rebuttal clarified that they are RL methods. Rebuttal also provide additional ablation without SFT. The rebuttal also provide various other ablations.

Reviewer rR7t asked about task generalization and the rebuttal clarified that experiments on it is already included. The rebuttal also provided additional comparison on 7B model and additional analysis on failures on different types of problems.

Reviewer Vx6G asked about how the reasoning change with budget and the rebuttal provide analysis of how reasoning is shortened. The reviewer is concerned that SFT is costly and the rebuttal clarified that SFT speeds later stage of training. The reviewer asked about comparison with BRPO and rebuttal provided an additional experiment.

Reviewer x15y commented that user would not know the right target length and the rebuttal limited the goal in the paper to hard constraint that are sometimes required.

**Reviewer Scores:**

Reviewer Nbim's concerns appear mostly answered, so the reviewer would likely retain or increase the score.
Reviewer rR7t 's concerns appear mostly answered, so the reviewer would likely retain or increase the score.
Reviewer Vx6G initially recommended reject. The rebuttal addressed some of the concerns, so the reviewer may possibly increase the score.
Reviewer x15y's concern is only partially addressed, so the reviewer would more likely retain the score.

---

### Decision · Program_Chairs · 2026-01-26

Reject